# Hydro-Servo-Aero-Elastic Analysis of Floating Offshore Wind Turbines

**Dimitris I. Manolas [1,2,*]** , **Vasilis A. Riziotis [1]**, **George P. Papadakis [3]** and **Spyros G. Voutsinas [1]**

[1] School of Mechanical Engineering, National Technical University of Athens,
GR15780 Zografos, Athens, Greece; vasilis@fluid.mech.ntua.gr (V.A.R.); spyros@fluid.mech.ntua.gr (S.G.V.)
[2] iWind Renewables PC, GR15344 Gerakas, Athens, Greece
[3] School of Naval Architecture and Marine Engineering, National Technical University of Athens,
GR15780 Zografos, Athens, Greece; papis@fluid.mech.ntua.gr
[*] Correspondence: manolasd@fluid.mech.ntua.gr; Tel.: +30-210-772-1097

**Abstract:** A fully coupled hydro-servo-aero-elastic simulator for the analysis of floating offshore wind turbines (FOWTs) is presented. All physical aspects are addressed, and the corresponding equations are concurrently solved within the same computational framework, taking into account the wind and wave excitations, the aerodynamic response of the rotor, the hydrodynamic response of the floater, the structural dynamics of the turbine-floater-mooring lines assembly and finally the control system of the wind turbine. The components of the complex multi-physics system of a FOWT interact with each other in an implicitly coupled manner leading to a holistic type of modeling. Different modeling options, of varying fidelity and computational cost, are made available with respect to rotor aerodynamics, hydrodynamic loading of the floater and mooring system dynamics that allow for timely routine certification simulations, but also for computationally intense simulations of less conventional operating states. Structural dynamics is based on nonlinear multibody analysis that allows reproducing the large rigid body motions undergone by the FOWT, as well as large deflections and rotations of the highly flexible blades. The paper includes the description of the main physical models, of the interaction and solution strategy and representative results. Verification is carried out by comparing with other state-of-art tools that participated in the Offshore Code Comparison Collaboration Continuation (OC4) IEA Annex, while the advanced simulation capabilities are demonstrated in the case of half-wake interaction of floating wind turbines by employing the free-wake aerodynamic method.

**Keywords:** wind energy; offshore wind turbines; floating wind turbines; multibody dynamics; finite element method (FEM) models; free-wake aerodynamics; blade element momentum models; coupled hydro-servo-aero-elastic analysis; hydrodynamic analysis; aeroelasticity

## 1. Introduction

Over the last decade, the advantages of offshore wind have increasingly attracted energy supply developers. Offshore sites exhibit substantially higher mean wind speeds and reduced levels of turbulence implying higher power production and lower fatigue loads, while public annoyance is much less than that for onshore installations since visual impact and noise are substantially lower. Over the last decade, the offshore installed capacity in Europe has increased from 2 GW in 2009 to 22 GW in 2019, which, in terms of the total installed wind capacity, corresponds to a change from 2.5% to 11%. It is thus, understood that meeting EU's target of a 300 GW overall installed capacity by 2030 will greatly rely on offshore wind energy development.

Today, the vast majority of the offshore wind farms are in shallow waters, utilizing support structures that are fixed in the seabed (bottom-fixed). However, as shallow water sites become scarce, the anticipated overwhelming development is expected to exploit sites of high wind potential in deep waters, with floating systems. Floating offshore wind turbines (FOWTs), whose development nowadays is in transition from demonstration to commercialization, face more challenging wind and wave excitations as compared to their bottom-fixed counterparts. Because of that, but also because of the increased cost of floating substructures, the overall cost of wind energy in deep waters is expected to be higher. Therefore, in order to maintain the competitiveness of wind energy, wind turbines need to be up-scaled leading to more delicate designs, with lightweight, flexible rotors. The designs of very big and highly flexible rotors that are mounted on (moving) floating substructures call for dedicated servo-hydro-aero-elastic tools formulated in a holistic, interdisciplinary context. In such an approach, all the components of a FOWT system are simulated simultaneously and are implicitly coupled within the same modeling environment, while all disciplines involved (i.e., aerodynamics, structural dynamics, elasticity, hydrodynamics and control) are modeled using the highest possible level of fidelity. In particular, holistic design tools for FOWT turbines should be able to account for: The complex, stochastic combined wind and wave excitation, the aerodynamics of a rotor undergoing large motions induced by the floating platform, the highly nonlinear structural dynamics of the flexible components (the blades, the shaft, the tower, the floater and the mooring lines), the hydrodynamic loading on complex platform geometries and finally the controls of the turbines. Moreover, the tools should be able to provide design loads for the subcomponents of the system for all possible design and environmental conditions that the turbine encounters during its lifetime. In this regard, the challenges of such all-inclusive modeling are next discussed in more detail also in reference to the current state-of-the art.

Aerodynamics: The modeling challenges include: The unsteady nature of the inflow due to ambient turbulence, wind shear, wind yaw misalignment and the floater induced motions (in particular the pitching and yawing motions); the onset of stall over the rotor blades, a crucial flow feature especially in standstill conditions; the effect of the 3D wake development and finally the nonlinear aeroelastic coupling. This is a quite demanding mix, far more complicated than in other applications, so that one would expect to find in use only advanced aerodynamic models. However, this is not the case. The vast majority of the existing state-of-the-art design and analysis tools are based on the blade element momentum theory (BEMT) [1]. Other more sophisticated models do exist in two main options; those based on free-wake potential methods [2] (medium fidelity) and those based on computational fluid dynamics (CFD) viscous solvers [3] (high fidelity). Both options have been utilized in FOWT hydro-servo-aeroelastic analyses but only to a limited extent, by only few research groups, while their application is mainly restricted to research rather than design work, aiming at a better understanding of the underlying physics and at assessing or calibrating BEMT-based models. It is noted that, although CFD models have been integrated into holistic modeling framework tools, due to their excessive computational cost they have only been employed in simulations of deterministic inflow conditions [3,4], or simplified turbine configurations in which critical flexibilities of the system (e.g., rotor and tower flexibilities) are suppressed, aiming at moderating the computational effort.

With regard to BEMT-based models, there exist various versions that differ in their details, some related to purely implementation aspects [5]. Although there is no complete consensus, still there is good agreement on the quality of predictions obtained with BEMT models when properly calibrated. Semi-empirical models, although not strictly predictive, are tunable and can indeed become very good design tools. Specifically to FOWT, care is needed in the application of the corrections that account for the platform induced dynamic tilt and yaw motions (combined yaw misalignment and flow inclination effects) [6].

The next alternative to BEMT is the free-wake potential models that apply vortex methods [7]. Vortex theory is quite old and is part of the classical aerodynamic theory. The most well-known examples are the lifting-line and lifting surface theories that were developed in the early 1960s for

fixed-wing aircraft. Vortex models are 3D by construction, with strong coupling along the span, which is completely absent in BEMT. In vortex models the tip and root corrections implemented in BEMT are no longer needed, while the assumption of infinite number of blades made by BEMT (in the momentum part) is dropped. Additionally, because of the free-wake formulation, the response to dynamic inflow is inherently accounted for. Although vortex methods address most of BEMT shortcomings, they still rely on 2D aerodynamic polars that a-posteriori correct the loads for viscous effects.

CFD viscous flow solvers are regarded as the best choice in terms of completeness and accuracy. The most frequent option is to solve the Reynolds-averaged Navier–Stokes (RANS) equations supplemented with appropriate turbulence closure and transition modeling. Out of the variety of turbulence closures, the most frequently applied are the eddy viscosity models like the k-ω [8,9]. Although still very expensive for aeroelastic simulations, eddy viscosity modeling is by far less expensive than the more advanced closures, detached eddy simulation (DES) or large eddy simulation (LES) [4,10].

Hydrodynamics: Two methods are widely used for estimating the hydrodynamic loading. The first is based on potential hydrodynamic theory and the second on Morison's empirical equation. Linear potential theory accounts for wave excitation, diffraction and radiation (added mass and added damping) [11] and is valid in case of small to moderate waves. Radiation concerns the six rigid body motions of the floater and should be extended to also include additional elastic modes in case the flexibility of the floater is considered. Potential theory can be extended to second order, in order to include mean drift forces or second order difference and/or sum frequency loads. These loads are proportional to the square of the wave amplitude A and could be of significant importance in case of increased wave steepness (H/λ) at low or high wave frequencies [12]. Often, only the difference frequency loads are considered using Newman's approximation [13], which only requires the mean drift forces from the solution of the first order hydrodynamic problem [14]. In order to account for the viscous damping, the quadratic damping term appearing in Morison's equation is usually added on top.

On the other hand, Morison's semi-empirical equation [15] applies to slender bodies that may be flexible and is valid in case the ratio $\lambda/D > 5$, where $D$ is the characteristic length of the body (i.e., the diameter) and $\lambda$ the wave length. Two hydrodynamic coefficients (inertia and drag coefficients) should be defined. Morison's equation accounts for wave excitation, diffraction, hydrodynamic added mass and viscous effects, while the linear radiation hydrodynamic damping is not accounted for. Since Morison's equation is subjected to calibration, a certain level of uncertainty is introduced. At engineering level, if the empirical parameters are fitted based on measured data, then higher order wave theories such as the stream function [16], can be used. Currently, for bottom mounted support structures, the hydrodynamic modeling is exclusively performed by Morison's equation, while for floating wind turbines both the potential theory and Morison's equation are applied.

In recent CFD based hydro-aero-elastic simulations, both rotor aerodynamics and hydrodynamic loads on the floater and the mooring lines are computed either simultaneously using the Volume of Fluid method [3], or via a hybrid approach in which the aerodynamic loads are computed using standard grid based Eulerian CFD analysis while hydrodynamic loads are accounted for using a Lagrangian smoothed particle hydrodynamics (SPHs) [4] approach. For the time being, this type of analysis costs a lot and is limited to simplified configurations and/or flow conditions.

Structural modeling: For wind turbines, beam theory is almost exclusively used [17]. Full 3D FEM structural modeling is only applied to certain areas such as the hub and the nacelle, where beam theory fails to provide design information. All early developments considered linear classical beam modeling. Making ad-hoc additions or corrections, certain nonlinearities were included, as, for example, the centrifugal stiffening due to the rotation of the blades. However, the basic modeling remained linear giving relatively stiff designs. In this context, linear models provided sufficiently good predictions, so there was no need to consider more advanced structural modeling. However, as the size of commercial turbines increased, and the pressure for cost reduction became critical, blades started to

become more flexible and the validity of linear beam theory was put in doubt. Larger deflections were anticipated leading to significant nonlinear coupling effects and potentially unfavorable blade loading [18].

Therefore, over the last ten years, a number of advanced nonlinear beam models have been implemented in the new generation of aeroelastic design tools. Options in this respect include nonlinear geometrically exact formulations (generalized Timoshenko methods) [19], multi-body formulations [20] and truncation methodologies [21]. Methods in the first category apply an extended form of Hamilton's variational principle. However, instead of expressing variations in terms of displacement and rotation variables, they adopt an intrinsic formulation of beam theory, see [19]. The formulation is intrinsic if not tied to a specific choice of displacement and rotation variables. Methods in the second category divide each component into a number of interconnected elements that are either considered as (flexible) beams or as rigid bodies. For the flexible beams, all types of models have been used, ranging from simple linear to geometrically exact and nonlinear ones. Finally, the last category comprises methods in which the nonlinear dynamic equations of the deformed beam are derived by following a consistent ordering scheme, whereby higher order terms beyond a certain degree of accuracy are truncated [22]. In deriving the dynamic equations, the assumption of homogenous isotropic material is usually made. In this way, the necessary input is limited to the standard set of structural properties (bending stiffness in flapwise and edgewise direction, torsional stiffness etc.). Recently a number of methods have been developed in which the full stiffness matrix of the composite lay-up is taken into consideration [23]. Methods of this type need, as input, the detailed inner structure of the composite materials in order to derive appropriate equivalent beam structural properties. Such models become necessary as passive load alleviation based on the structural tailoring of the inner structure of the blades [23,24] is gaining attention.

Amongst the existing design tools, some directly apply the finite element method (FEM) to the full dynamic system of equations [18], while others adopt reduced order modeling based on various order reduction techniques, such as the linear modal expansion [1] or the Craig-Bampton method [25]. In the latter case, the aim is to substantially reduce computational effort in view of the long list of simulations required by the IEC standard [26].

Mooring lines: The modeling is defined either in quasi-static or fully dynamic context and discretized using FEM. In every time step, quasi-static models solve the static catenary equations without considering inertial effects or hydrodynamic loading (see for example [27]). On the contrary, dynamic models solve the dynamic equilibrium equations of the mooring lines, as performed for any other flexible part of the wind turbine. Inertia of the mooring lines and gravity load, as well as hydrodynamic loads (by means of Morison's equation), are usually included. The interaction of the mooring line with the seabed is important and is usually modeled by non-linear springs and dampers along the seabed that prevent the catenary chain to be further submerged.

Hydro-servo-aero-elastic synthetic modeling tools are classified depending on the methods adopted for the modeling of each module (hydro, servo, aero, elastic), as detailed in the previous paragraphs. The hGAST model that is presented in the paper combines different modeling options of varying fidelity for aerodynamic and hydrodynamic analysis. Rotor aerodynamics are handled by either an engineering BEMT model enhanced with various corrections to account for effects that lie beyond the capabilities of momentum theory or a medium fidelity free-wake vortex model. In a similar manner, hydrodynamic loads can be either calculated through the low fidelity Morison's equation or by means of medium fidelity potential theory. The low fidelity options are mainly used for routine certification and design simulations of mature concepts at conventional operating conditions. The medium fidelity options are selected in case novel concepts or less conventional operational states are addressed and deeper physical insight is required. Selection of the sub-models in such a holistic context balances between accuracy and cost. The least possible loss of fidelity is introduced by the default option, while special care has been taken in reducing the run time when using costly options (e.g., particle mesh acceleration technique in vortex simulations). The structural dynamics

modeling employed in the model is the state-of-the-art for this type of analyses. It allows for large rigid body motions and large deflections and rotations of highly flexible components of the wind turbine (e.g., the blades) or of the mooring lines. Finally, in the context of the generalized modeling environment any aerodynamic or hydrodynamic sub-module can be easily plugged into the structural module without cancelling the nonlinear character of the underlying interactions.

The backbone of the solver is based on the multibody methodology implemented in the General Aeroelastic Structural Tool (GAST), developed in the 1990s [28], which over the years went through several revisions and extensions. In its present form, the hydro-servo-aero-elastic tool offers advanced features lying at or beyond the current state-of-the-art employed in most of the existing software used either for academic or industrial purposes. More specifically, geometric non-linear structural and inertial effects due to large deflections and rotations can be captured within the multibody structural dynamic modeling context [18]. Moreover, prebend and/or swept blades can be consistently modeled in the structural dynamics and the aerodynamic modules, respectively [29,30]. In addition, fully populated stiffness properties can be introduced in the beam modeling that account for the anisotropy of the material which is of particular importance when simulating blades with bend twist coupling (BTC) capabilities [23]. Another advantage of the dynamic modeling approach considered in the model is that aeroelastic equations are derived in their linearized formed which allows solving the linear stability problem for the full configuration at any highly deflected state [18]. Both linear eigenvalue [31] and nonlinear [32] aeroelastic stability analysis can be performed, taking into account the control equations in normal operation (closed loop analysis) [33] or in the idling state in storm conditions [34]. Finally, to the best of the authors' knowledge, the present tool is the only existing in the wind energy sector that is coupled to an efficient, free-wake aerodynamic solver [2,7,30] combined with a multi-block particle mesh technique [35], based on the algorithm presented in [36]. This combination drastically reduces the computational time of the particle-to-particle calculations needed for the evolution of the wake and renders simulations with millions of particles affordable. Through the above, simulation of certification design cases of ten minutes duration are made possible.

In the present work, the fully coupled hydro-servo-aero-elastic simulator for the analysis of floating offshore wind turbines, hGAST, is formulated and the theory of the various interacting modules is presented. Because there are no full-scale measured data available in open literature, proper validation is not possible. This led the Wind Energy community to organize code-to-code comparisons, of which the most widely referenced are in the series of the OC IEA Annexes [37] coordinated by NREL. Though, verification of the code is carried out by comparing predictions against other participating state-of-art tools results. The full configuration is considered with concurrent wind and wave exciting, the controller in closed loop operation and all the structural flexibilities enabled. Then in order to demonstrate the simulation capabilities of the tool, the free-wake vortex aerodynamic method in employed and representative results from the half-wake interaction problem (rotor interacts with the wake of an upstream turbine in partial shading state) for floating wind turbines are discussed. It is worth noting that the simulation of 300 s that considers both rotors was performed on a PC with 6 CPUs in two days.

## 2. The Coupled Simulation Methodology

Fully coupled hydro-servo-aero-elastic simulations of offshore wind turbines can be formulated within the framework of dynamic systems. The complete multi-physics system is considered as a combination of interacting subcomponents interpreted in a generalized context. The overall concept is illustrated in Figure 1 and includes:

- The turbulent wind inflow that interacts with all parts of the structure above the sea level;
- The incoming sea waves and currents that interact with the submerged fixed or floating support structure;
- The structure of the turbine that interacts on one hand with the air and water flows and on the other with the generator (and eventually the electrical grid) and the control system;

- The structure of the seabed that interacts with the foundation of the mooring lines;
- The control system which specifies the pitch for every blade and the generator torque based on response input and by that interacts with the structure;
- The electrical system which provides the generator's operating condition.

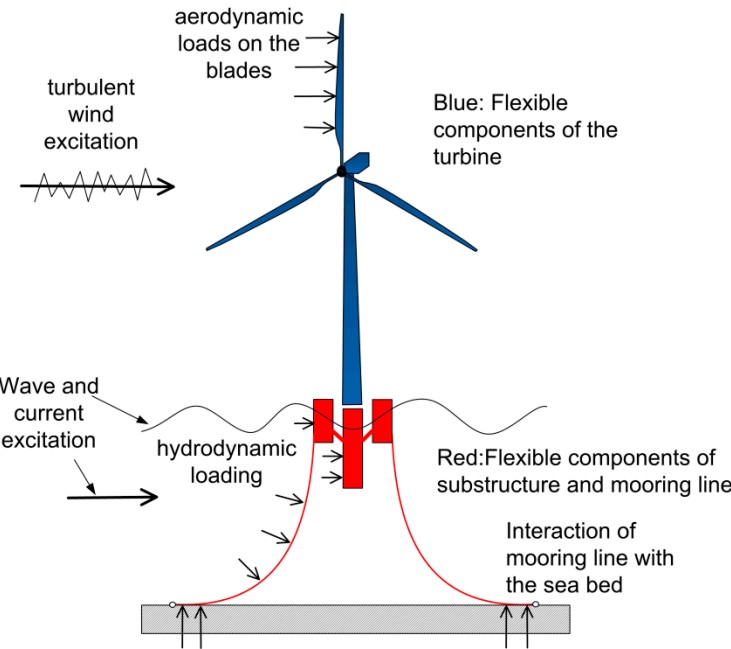

**Figure 1.** Sub-components of the multi-physics floating offshore wind turbine (FOWT) system.

### 2.1. Structural (Solid) Dynamics Modeling

The solid parts of a FOWT include the turbine, the floater and the mooring lines as subcomponents. Multibody dynamics is adopted in the formulation; all flexible components are represented as 1D (beam) structures, whereas numerical discretization applies nonlinear FEM. Multibody dynamics can concurrently accommodate rigid body motions and structural flexibility together with the nonlinear inertial and structural effects and geometric couplings that large deflections of highly flexible components may generate (e.g., blades). Rigid motions are imposed by either the control system (e.g., blades' pitch motion, drive train rotational speed) or by external excitation (e.g., floater motions due to waves and blades rotation due to wind).

#### 2.1.1. Wind Turbine and Floater

The solid components of an offshore wind turbine configuration include the members of the support structure (the buoyant elements and brackets of the floater or struts of the bottom-fixed support), the blades, the drive train and the tower. All the above flexible components, which are usually sufficiently elongated, are approximated as a collection of linear Timoshenko beam structures. Other three dimensional structures, such as the hub, the generator, etc., are considered rigid and modeled as lumped masses to which flexibility may be added in the form of a concentrated spring (e.g., the yaw mechanism).

In multibody dynamics, assembly of the components into the full system is carried out by imposing appropriate kinematic and loading conditions at the connection points. In brief, at any connection point, one of the connected component specifies the position (displacements) and orientation (rotations), while all others contribute their reaction loads [18].

A local coordinate system $[O, x, y, z]$ (see Figure 2a) is assigned to every component (body) with respect to which local elastic displacements are defined. The local frame of every body (e.g.,

blades, shaft, tower, floater buoys) is subjected to rigid body motions, each defined by a 6-component vector, including 3 translations and 3 rotations. There are rigid body motions that bodies undergo by themselves and motions that are induced through their connections to other bodies. For the $k^{th}$ body, let $\mathbf{q}^k = \left\{ \ \mathbf{q}_t^k, \ \ \mathbf{q}_r^k \ \right\}^T$ denote the set of time invariant displacements and rotations and time variant body self-motions that define the origin and the orientation of the body local system with respect to the inertial frame in the un-deflected state, and $\mathbf{q}_0^k = \left\{ \ \mathbf{q}_{0t}^k, \ \ \mathbf{q}_{0r}^k \ \right\}^T$ denote the motions due to connections. Rigid translations (denoted by sub-script "t") will displace the body as a whole to positions denoted by $\mathbf{R}^k$ and $\mathbf{R}_0^k$, respectively, for the two types of motions, while rigid rotations (denoted by sub-script "r") lead to rotation matrices $\mathbf{T}^k$ and $\mathbf{T}_0^k$. Based on the above, the (global) position vector $\mathbf{r}_G^k$ of any arbitrary point P over the deflected $k^{th}$ body with respect to the global inertia frame $\left[ O_G, x_G, y_G, z_G \right]$ (see Figure 2a) is expressed through the (local) position vector of P in the un-deflected state $\mathbf{r}_l^k = (x_l^k, y_l^k, z_l^k)^T$ and the vector of the local elastic deflections (displacements and rotations) $\boldsymbol{u}^k = \left( u^k, v^k, w^k, \theta_x^k, \theta_y^k, \theta_z^k \right)^T$ (both defined in body local co-ordinates), and the motions $\mathbf{q}^k$ and $\mathbf{q}_0^k$ as follows:

$$\mathbf{r}_G^k = \mathbf{R}^k(\mathbf{q}^k; t) + \mathbf{T}^k(\mathbf{q}_r^k; t) \cdot \left( \mathbf{R}_0^k(\mathbf{q}_0^k; t) + \mathbf{T}_0^k(\mathbf{q}_{0r}^k; t) \cdot \left( \mathbf{r}_l^k + \mathbf{S} \cdot \boldsymbol{u}^k(t) \right) \right) \tag{1}$$

where $S$ matrix is given by:

$$S = \begin{bmatrix} 1 & 0 & 0 & 0 & z_l^k & 0 \\ 0 & 1 & 0 & -z_l^k & 0 & x_l^k \\ 0 & 0 & 1 & 0 & -x_l^k & 0 \end{bmatrix} \tag{2}$$

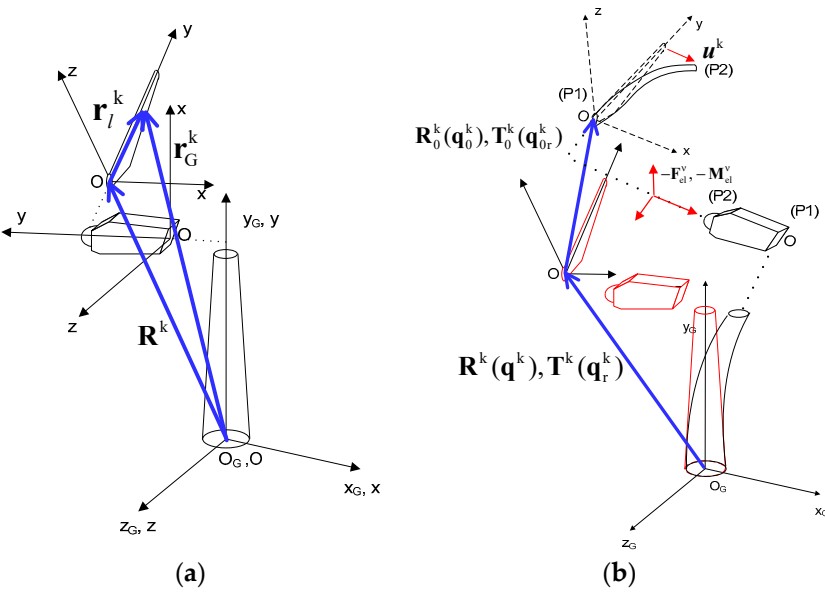

(**a**)            (**b**)

**Figure 2.** (**a**) Local co-ordinate systems of the different flexible bodies forming the wind turbine system, (**b**) application of multibody kinematics at the turbine level (connected bodies).

In Equation (1), $\mathbf{q}^k$ convert local to global co-ordinates (and vice versa) through $\mathbf{R}^k$ and $\mathbf{T}^k$ and does not include degrees of freedom of rigid body or elastic motion communicated by other bodies connected with k body. These motions are included in $\mathbf{q}_0^k$ and appear in (1) through $\mathbf{R}_0^k$ and $\mathbf{T}_0^k$. In Figure 2b, $\mathbf{q}_0^k$ concerns the motions that are induced on the blade by the end of the drive train in the deflected state, while $\mathbf{q}^k$ concerns the fixed offset and rotations of the blade local system with respect to the inertial frame plus the pitch motion of the blade (body self-motion).

Kinematic constraints are applied to the end nodes of connected bodies (P1 and P2 in Figure 2b). For example, node P1 of every blade is rigidly connected to node P2 of the drive train. In this case, the rigid connection kinematic conditions are expressed through the nonlinear velocity/angular velocity equations:

$$\dot{\mathbf{r}}_G^{blade}(P1) = \dot{\mathbf{r}}_G^{d\_train}(P2)$$
$$\dot{\boldsymbol{\theta}}_G^{blade}(P1) = \dot{\boldsymbol{\theta}}_G^{d\_train}(P2)$$

(3)

where $\dot{\boldsymbol{\theta}}_G = \left(\dot{\theta}_{G1}, \dot{\theta}_{G2}, \dot{\theta}_{G3}\right)^T$ is the angular velocity vector of a body's end node (P1 or P2) with respect to the inertial frame. While linear velocities $\dot{\mathbf{r}}_G$ are derived through differentiation of (1), angular velocities are obtained through:

$$\left(\mathbf{T}^k \cdot \mathbf{T}_0^k\right)^T \cdot \left(\mathbf{T}^k \cdot \mathbf{T}_0^k\right) = \begin{pmatrix} 0 & -\dot{\theta}_{l3}^k & \dot{\theta}_{l2}^k \\ \dot{\theta}_{lz}^k & 0 & -\dot{\theta}_{l1}^k \\ -\dot{\theta}_{l2}^k & \dot{\theta}_{l1}^k & 0 \end{pmatrix}$$

(4)

where $\dot{\boldsymbol{\theta}}_l^k = \left(\dot{\theta}_{l1}^k, \dot{\theta}_{l2}^k, \dot{\theta}_{l3}^k\right)^T$ is the angular velocity vector of the $k^{th}$ body expressed with respect to its local co-ordinate system. Then,

$$\dot{\boldsymbol{\theta}}_G^k = \left(\mathbf{T}^k \cdot \mathbf{T}_0^k\right) \cdot \dot{\boldsymbol{\theta}}_l^k$$

(5)

In addition to the kinematic conditions that are imposed at the connection points, loading conditions must be also satisfied. In particular, at every connection point, one of the connected bodies contributes the displacements and rotations to the others, which in turn contribute their internal (reaction) loads. In the above example, the blade receives displacements and rotations from the drive train end node P2, while the drive train end node receives reaction loads of node P1 of the blade. It is noted that reaction loads of a body can be expressed as functions of the degrees of freedom of the body and; therefore, an implicit coupling between bodies can be readily established.

Of particular importance is that the above multibody formulation is extended to the body level. Highly flexible bodies, such as the blades, are divided into a number of interconnected sub-bodies, each considered as a single or as an assembly of linear beam elements. Large deflections and rotations gradually build up and nonlinear dynamics are introduced by imposing to each sub-body, the deflections and rotations of preceding sub-bodies as rigid body motions. Dynamic coupling of the sub-bodies is again introduced by communicating the reaction loads (three forces and three moments), at the first node of every sub-body to the free node of the previous sub-body, as external load. Illustration of the extension of the multibody framework at the level of a body is illustrated in Figure 3. Linear Timoshenko beam modeling is applied in order to account for the local deflections vector $\boldsymbol{u}^k$ of every flexible body/sub-body. The coordinate system $[O', \xi, \eta, \zeta]$ shown in Figure 4 is the cross section local system. Around the axes of this system, local bending and torsion rotations take place. The set of dynamic equilibrium equations (three force and three moment equations) of the $k^{th}$ body takes the form:

$$\int_A \rho dA \left(\mathbf{S}^T \cdot \mathbf{S}\right) \cdot \left(\mathbf{T}^k \cdot \mathbf{T}_0^k\right)^T \cdot \ddot{\mathbf{r}}_G^k = \begin{pmatrix} F'_x \\ F'_y \\ F'_z \\ M'_x + F_z - F_y w'^k \\ M'_y \\ M'_z - F_x + F_y u'^k \end{pmatrix}^k + \begin{array}{c} \text{external loads including} \\ \text{reaction loads communicated} \\ \text{by connected bodies} \end{array}$$

(6)

where the prime symbol (′) denotes derivatives with respect to the beamwise local y direction. The terms $F_y w'^k$ and $F_y u'^k$ in the moment x and z equations are the only non-linear terms considered

in the analysis. This is because they are expected to contribute significantly, especially in the case of rotating beams in which axial force increases due to the centrifugal effect. The two terms give rise to virtual stiffening of the beam as rotational speed increases.

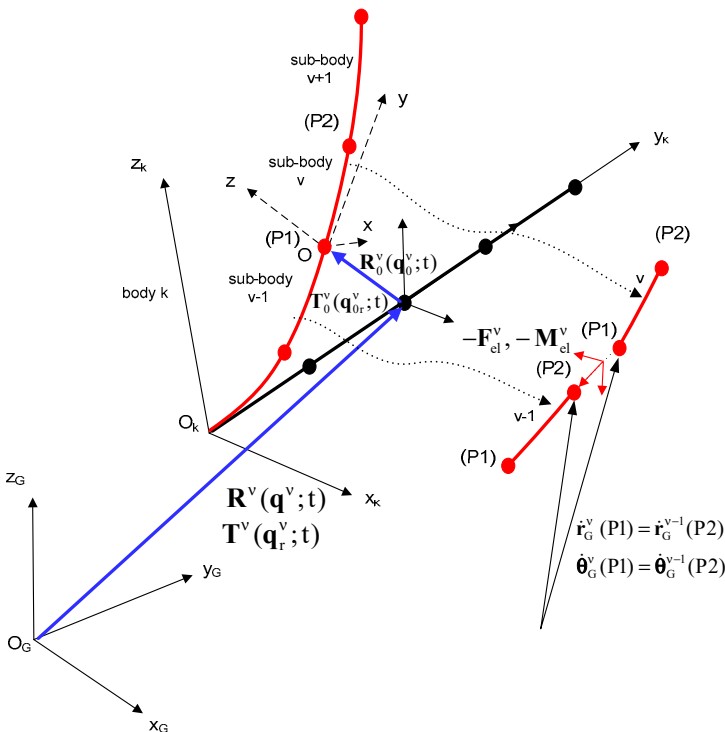

**Figure 3.** Application of multibody kinematics at the body level (connected sub-bodies).

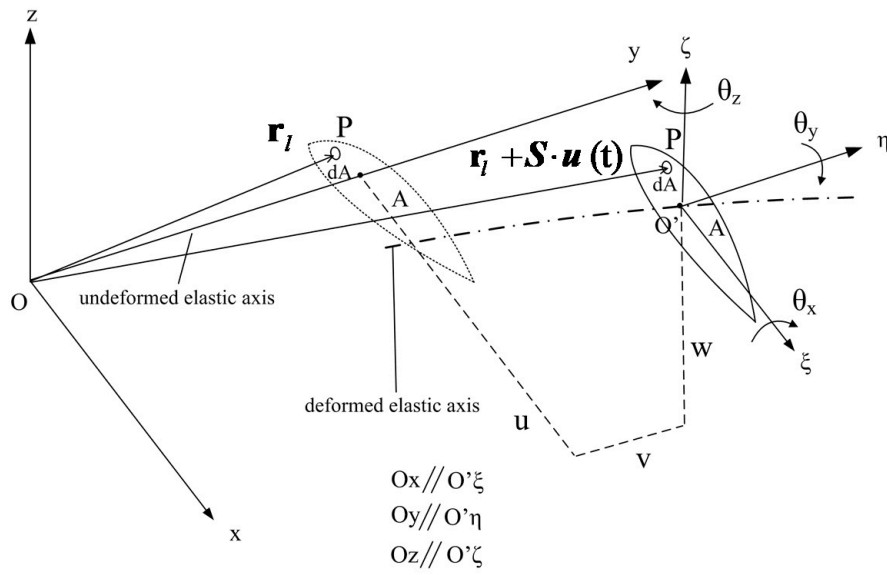

**Figure 4.** Definition of body local deflections.

The constitutive relation between the generalized structural loads over a cross section of a beam and the strains-curvatures is given by:

$$\mathbf{F}^k = \mathbf{K}^k \cdot \boldsymbol{\varepsilon}^k, \ \mathbf{F}^k = \left( F_x^k, F_y^k, F_z^k, M_x^k, M_y^k, M_z^k \right)^T, \ \boldsymbol{\varepsilon}^k = \left( \gamma_x^k, \varepsilon_y^k, \gamma_z^k, k_x^k, k_y^k, k_z^k \right)^T \qquad (7)$$

where,

$$
\mathbf{K}^k = \left\{ \begin{matrix}
K_{xx}^A & K_{xy}^A & K_{xz}^A & K_{xx}^B & K_{xy}^B & K_{xz}^B \\
 & K_{yy}^A & K_{yz}^A & K_{yx}^B & K_{yy}^B & K_{yz}^B \\
 & & K_{zz}^A & K_{zx}^B & K_{zy}^B & K_{zz}^B \\
 & \text{sym.} & & k_{xx}^C & k_{xy}^C & k_{xz}^C \\
 & & & & k_{yy}^C & k_{yz}^C \\
 & & & & & k_{zz}^C
\end{matrix} \right\}^k
\tag{8}
$$

and,

$$
\varepsilon_y = v', \ \gamma_x = u' + \theta_z, \ \gamma_z = w' - \theta_x, \ k_x = \theta'_x, \ k_y = \theta'_y, \ k_z = \theta'_z \text{ for the } k-\text{th Body} \tag{9}
$$

In (8), $\mathbf{K}^k$ is the sectional Timoshenko full stiffness matrix. With respect to a standard Timoshenko beam approach, the elements $K_{xx}^A$ and $K_{zz}^A$ represent transverse shear rigidity, $K_{yy}^A$ axial stiffness, $K_{xx}^C$ and $K_{zz}^C$ flexural stiffness in flap- and edge-wise directions, respectively, and $K_{yy}^C$ torsional stiffness. The off-diagonal elements $K_{xy}^C$ and $K_{yz}^C$ are responsible for the activation of bend-twist coupling. By introducing (9) into (7), $\mathbf{K}^k$ can be split as follows:

$$
\mathbf{F}^k = \mathbf{K}^k \cdot \varepsilon^k = \mathbf{K}_1^k \boldsymbol{u'}^k + \mathbf{K}_2^k \boldsymbol{u}^k \tag{10}
$$

By substituting (10) into (6), the dynamic equations are expressed with respect to the considered degrees of freedom in the form:

$$
\int_A \rho dA \left( \mathbf{S}^T \cdot \mathbf{S} \right) \cdot \left( \mathbf{T}^k \cdot \mathbf{T}_0^k \right)^T \cdot \ddot{\mathbf{r}}_G^k = \left( \mathbf{K}_1^k \boldsymbol{u'}^k \right)' + \left( \mathbf{K}_2^k \boldsymbol{u}^k \right)' + \left( \mathbf{K}_3^k \boldsymbol{u'}^k \right) + \left( \mathbf{K}_4^k \boldsymbol{u}^k \right) + \text{ external loads} \tag{11}
$$

In order to apply the FEM, the above set is written in weak form by applying the principle of virtual work and then discretized using first-order shape functions for extension and torsion, and modified $C^1$ Hermitian functions are used for the two bending displacements that prevent shear locking effect [38] by satisfying static equilibrium compatibility relations.

The discrete FEM Equation (11) and the constraint Equation (3) are analytically linearized and expressed in perturbed form. The advantage of the proposed formulation, in comparison to other multibody formulations applying the Lagrange multipliers approach, is that linear eigenvalue stability analysis can be performed in the resulting linearized dynamic system with respect to a highly deflected steady or periodic state. Unknowns of the structural dynamic problem are the $\mathbf{q}^k$ and $\mathbf{q}_0^k$ vectors, defined through (i) kinematic constraint equations (like those given by Equation (3)) or (ii) proper control and dynamic equations (e.g., pitch control equation, drive train equation and floater dynamic equations) and the vector of the body local discrete elastic degrees of freedom $\boldsymbol{u}^k = \left( u^k, v^k, w^k, \theta_x^k, \theta_y^k, \theta_z^k \right)^T$.

### 2.1.2. Mooring Lines

Modeling of the mooring line is based on 1D nonlinear FEM. The mooring lines are modeled as space truss elements that only transfer tensile forces while their compression stiffness is zero [39]. Each element is subjected to inertial, gravitational and hydrodynamic loading, calculated by means of Morison's equation. The seabed interaction is modeled by appropriate stiffness and damping terms. Coupling of the mooring lines with the floater is established by imposing as kinematic constraint conditions the motions of the floater to the mooring line node attached to the floater, while the mooring line returns reaction loads to the floater. Both catenary mooring lines and tendons for tension leg platforms (TLP) can be modeled.

In Figure 5, a truss element $P_0 Q_0$ of initial undeformed length $\ell_0 = 2\alpha_0$ is shown. The non-dimensional coordinate $\xi$ (ranging from $-1$ to $1$) is used to define the arbitrary point

$R_0$ along the element. The position vector of this point with respect to the inertial frame is initially $r_0$. As a result of rigid body motion and elastic deformation, the truss element will move to a new position $P_nQ_n$. The arbitrary point along the element (point $R_n$) will now have a position vector $r_n$, given by the following expression:

$$\mathbf{r}_n = \mathbf{r}_0 + \boldsymbol{u} \tag{12}$$

where vector $\boldsymbol{u}$ represents displacement due to elastic deflection and rigid body motion of the element.

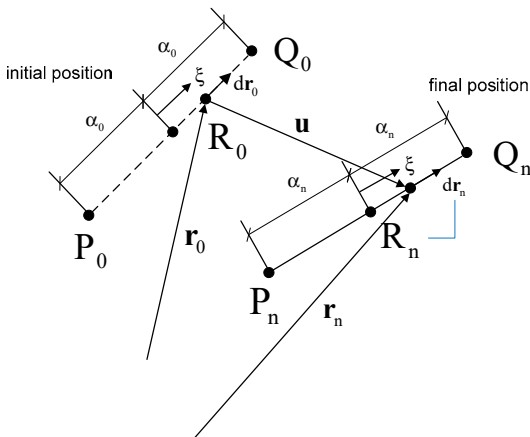

**Figure 5.** Truss element in undeformed and deformed states.

By applying the virtual work principle, the dynamic equations of a truss element in space are obtained. The principle requires that, for an arbitrary virtual displacement field, $\delta\boldsymbol{u}$:

$$\int_{\ell_0} \delta\boldsymbol{u}^T \mathbf{q}^i \, d\ell = -\int_{V_0} \sigma \, \delta\varepsilon \, dV + \int_{\ell_0} \delta\boldsymbol{u}^T \mathbf{q}^e \, d\ell \tag{13}$$

where $\mathbf{q}^i$ and $\mathbf{q}^e$ are the inertial and external (hydrodynamic, gravitational and buoyancy) per unit length forces acting on the element, $\sigma$ and $\delta\varepsilon$ are the axial stress and the incremental variation of the axial strain caused by the relative motion of the element ends P and Q, and $V_0$ the volume of the undeformed element. The standard definition of the engineering strain [40],

$$\varepsilon = \frac{\ell_n - \ell_0}{\ell_0} \tag{14}$$

is used in order to define the variation $\delta\varepsilon$. By considering that the displacements of the two end nodes, P and Q, of the element are the degrees of freedom of the problem, the displacement of any intermediate point along the element is written as

$$\boldsymbol{u} = \mathbf{N} \cdot \hat{\boldsymbol{u}}, \ \hat{\boldsymbol{u}} = \left\{ \underbrace{u_1 \ v_1 \ w_1}_{\text{node P}} \ \underbrace{u_2 \ v_2 \ w_2}_{\text{node Q}} \right\}^T \tag{15}$$

where $\mathbf{N}$ is the matrix of the shape functions considered linear in the present analysis.

Using the same linear interpolation functions, the coordinates of the arbitrary point $R_0$ along the undeformed element can be defined in terms of the end points $P_0\left(x_1, y_1, z_1\right)$ and $Q_0\left(x_2, y_2, z_2\right)$:

$$\mathbf{r}_0 = \mathbf{N} \cdot \hat{\mathbf{x}}, \ \hat{\mathbf{x}} = \left\{ x_1 \ \ y_1 \ \ z_1 \ \ x_2 \ \ y_2 \ \ z_2 \right\}^T \tag{16}$$

Using (14), (15) and (16), the incremental variation of the strain $\delta\varepsilon$ can be expressed in terms of the virtual displacement $\delta\hat{u}$ of the element end nodes:

$$\delta\varepsilon = \frac{1}{\ell_0}\mathbf{t}^{\mathrm{T}}\cdot\mathbf{A}\cdot\delta\hat{u} = \mathbf{b}^{T}\cdot\delta\hat{u} \tag{17}$$

where $\mathbf{t}$ is the unit vector tangent to the element and $\mathbf{A} = [-\mathrm{I}_{3\times3}, \mathrm{I}_{3\times3}]$ and $\mathrm{I}_{3\times3}$ the $3\times3$ unit matrix.

The stress field is determined through application of Hooke's laws by making the assumption of linear elastic material. As already stated, the Young modules $E$ of the mooring line will be different in compression and extension. The compression value will be equal to zero but in practical terms is taken very small, for numerical stability.

By introducing (17) into (13), and after integration over the cross section area $\mathrm{A}_0$, the following system of non-linear dynamic equations is obtained:

$$\int\limits_{-1}^{1}\mathbf{N}^{\mathrm{T}}\mathbf{q^e}\,\alpha_0\,\mathrm{d}\xi - \int\limits_{-1}^{1}\mathbf{N}^{\mathrm{T}}(\rho\mathrm{A}_0)\,\ddot{\mathbf{r}}_{\mathrm{n}}\,\alpha_0\,\mathrm{d}\xi = \int\limits_{-1}^{1}\mathbf{b}\,\sigma\,\mathrm{A}_0\,\alpha_0\,\mathrm{d}\xi = 2\alpha_0\mathrm{A}_0\,\mathbf{b}\,\sigma = \mathbf{f} \tag{18}$$

where $(\rho\mathrm{A}_0)$ is the linear mass distribution of the mooring line. The non-linearity of Equation (18) lies in two distinct reasons. The first is the inherent capability of the model to accommodate both large displacements of the truss elements and deformations (geometric non-linearity). The second is due to the different stiffness characteristics of the elements in tension and compression (material non-linearity).

The non-linear Equation (18) that concerns one element are assembled for all mooring elements. Then they are expressed in linear perturbed form (linearization about a reference deflected state) and added together with the rest of the structural equations. This final step is done by imposing connection conditions at the attachment points of the mooring lines on the floater. As shown in Figure 6, the floater defines the motions of the mooring line at the attachment point, while the mooring line gives back the tensile reaction force.

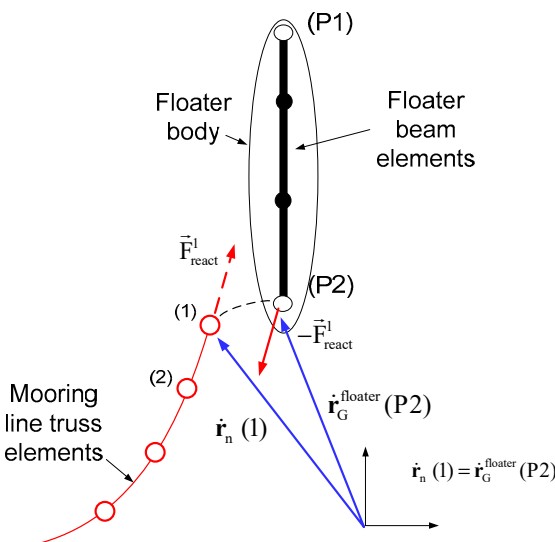

**Figure 6.** Kinematic and dynamic coupling of the turbine and the floater with the mooring line.

## 2.2. Aerodynamic Modeling

There are two options available for the modeling of the rotor aerodynamics: a low-cost and semi empirical enhanced model based on blade element momentum theory (BEMT) and a medium-cost and fidelity model based on free-wake vortex (VRX) modeling. The first is mainly used for certification

routine simulations, while the second is mainly selected when less conventional conditions with strong 3D effects, that lie out of the modeling capabilities of BEMT, are dealt with (e.g., high yaw misalignment, large floater motions, partial wake operation).

### 2.2.1. Blade Element Momentum Theory Modeling

The standard BEMT model is enhanced in three aspects: (a) a wake-skewed correction model that accounts for yaw and tilt misalignment [41], (b) a dynamic inflow model based on cylindrical wake modeling [42], (c) unsteady aerodynamics and dynamic stall modeling based on the ONERA [43] or the Beddoes–Leishman [44] engineering models. More details on the enhanced BEMT implemented in the model can be found in [45].

### 2.2.2. Free-Wake Model

The model considers the unsteady flow of an incompressible and inviscid fluid around a multi-component configuration of bodies that may move independently from each other. Thus, in addition to the isolated turbine problem, the case of more than one wind turbine in a cluster can be dealt with.

As illustrated in Figure 7, aerodynamic bodies (e.g., rotating blades) can be considered as (i) lifting lines, (ii) thin lifting surfaces and (iii) thick panel bodies. All aerodynamic bodies are divided into a number of "cross sections", which are perpendicular to the body reference line. Furthermore, a number of nodal points are defined over every cross section. The wing element between two consecutive sections forms the "strip". The strip is the main aerodynamic entity of the body, on which the local aerodynamic loads are calculated. In the lifting line analysis, the building block of the numerical grid is the strip. Each section has only two nodes describing the leading and trailing edge of the airfoil, which; therefore, collapses into a single flat plate element (panel) formed upon every strip, as shown in Figure 7. In the lifting surface and thick panel analysis, the building block of the numerical grid is the surface panel element formed by four grid nodes (two pairs belonging to consecutive sections) in clockwise direction. In this case, a number of surface elements (panels) form the strip, as shown in Figure 7. Within every panel, a single control point (collocation point) is defined, on which the circulation, the pressure jump or the local pressure is calculated for lifting line bodies, thin bodies or thick bodies, respectively.

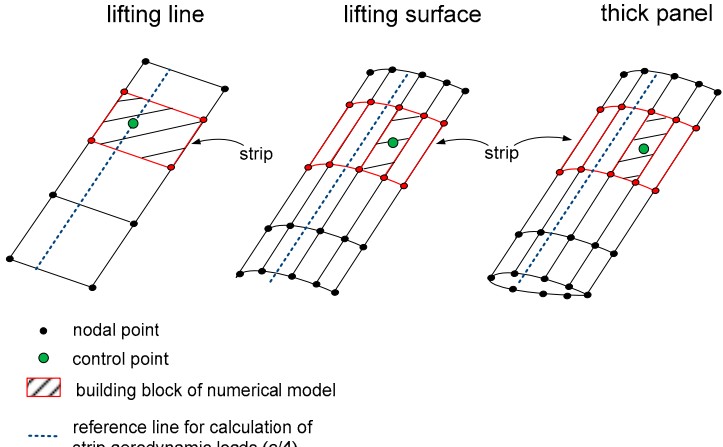

**Figure 7.** Aerodynamic modeling options of lifting bodies.

The theoretical backbone of the method is Helmholtz's decomposition theorem [46], according to which the flow field $\vec{u}$ at a point in space $\vec{x}$ can be decomposed as:

$$\vec{u}(\vec{x};t) = \vec{u}_{ext}(\vec{x};t) + \vec{u}_{solid}(\vec{x};t) + \vec{u}_{wake}(\vec{x};t) \tag{19}$$

where $\vec{u}_{solid} = \nabla \phi$ is the potential part of the flow associated with the presence of the solid boundaries (e.g., blades) in the flow, $\vec{u}_{wake} = \nabla \times \vec{\psi}$ is the vortical part associated with the wake or the free vorticity given by $\vec{\omega}$ and $\vec{u}_{ext}$ is a given external flow field. In the above $\phi$, $\vec{\psi}$ denote the scalar and vector potential of the flow, respectively, that satisfy the following field equations:

$$\nabla^2 \phi = \nabla \cdot \vec{u} = 0, \ \nabla^2 \vec{\psi} = -\nabla \times \vec{u} \equiv -\vec{\omega} \tag{20}$$

According to Green's theorem, $\vec{u}_{solid}$ can be expressed through surface singularity distributions. In the lifting line approach, the body contribution is considered through open "horseshoe" vortex filaments distributed over the strips. In the lifting surface and thick panel options, the body contribution is considered through surface dipole and surface dipole plus source distributions respectively, over the elements. It is noted that for any dipole surface distribution an equivalent line and surface vorticity distribution can be defined. Therefore, a general expression of $\vec{u}_{solid}$ is given by:

$$\vec{u}_{solid}(\vec{x};t) = \int_{S(t)} \frac{(\sigma \cdot + \vec{\gamma} \times) \vec{r}}{4\pi \vec{r}^3} dS(\vec{y}) \tag{21}$$

where, $\vec{r} = \vec{x} - \vec{y}$, $S(t)$ denotes collectively the solid boundaries and $\sigma$, $\vec{\gamma}$ denote the surface source and vorticity distributions. Similarly according to Green's theorem, $\vec{u}_{wake}$ can be expressed in the following integral form,

$$\vec{u}_{wake}(\vec{x};t) = \int_{D_\omega(t)} \frac{\vec{\omega}(\vec{y};t) \times \vec{r}}{4\pi \vec{r}^3} dD(\vec{y}) \tag{22}$$

where $D_\omega(t)$ denotes the region covered by the wake.

In the lifting line option, the bound line vorticity (equivalent to circulation around filaments), $\vec{\Gamma}$, of every strip is obtained from the solution of the discretized system of the nonlinear lifting line equations:

$$\vec{L}(\vec{x}_0;t) = \rho \vec{u}(\vec{x}_0;t) \times \vec{\Gamma}(\vec{x}_0;t) \tag{23}$$

satisfied on every control point $\vec{x}_0$ of the lifting line grid (i.e., the single control point per strip located at the quarter chord line in the midspan position of the strip). This solution uses as input the airfoil 2D aerodynamic polars at spanwise sections, while corrections to account for viscous and dynamic stall effects are applied based on the ONERA [43] dynamic stall model.

In the thin lifting surface and thick panel body options, the non-penetration, kinematic boundary condition is satisfied at the control points. In this case, the central point of every surface panel element. Furthermore, the Kutta condition is satisfied along the trailing edge (TE) line of all lifting bodies. In the above two modeling options, airfoil polars are not needed, except when applying viscous corrections. As in the lifting line case, this is done on an a-posteriori basis, using the ONERA dynamic stall model.

In potential theory, wakes are introduced as material surfaces carrying surface vorticity $\vec{\gamma}$. Existing wake models differ on the choice of elements to describe the wake, but most importantly on whether they assume connectivity among these elements. Connectivity is necessary in order to satisfy the fundamental requirement that vorticity is div free or else that the vorticity lines in the wake are either closed or they start and end on a solid boundary or at infinity. These requirements are, by construction, fulfilled when the wake is formed by means of vortex filaments or is retained as a surface. Connectivity; however, can generate numerical problems when the wake is excessively deformed or interacts with solid bodies, as in the present case. In order to bypass this problem, the present model uses freely moving vortex blobs. They are 3D point vortices equipped with core. Vortex blobs are generated at every time step in a two-step procedure. First the wake is released

in the form of surface vorticity. At this step, the no-penetration and Kutta conditions are satisfied. Then the convection step is carried out, during which surface vorticity is integrated and transformed into vortex blobs defined by their intensities $\vec{\Omega}_P$, positions $\vec{Z}_P$ and core sizes $\varepsilon$, as shown in Figure 8. The free vorticity is defined as follows:

$$\vec{\omega}(\vec{x};t) \cong \sum_P \vec{\Omega}_P(t)\, \zeta_\varepsilon(\vec{x} - \vec{Z}_P(t)) \tag{24}$$

where $\zeta_\varepsilon$ is the cut-off or distribution function defined over the core. In the current implementation, a cubic exponential function [47] is used, which results in the following velocity representation:

$$\vec{u}_{wake}(\vec{x};t) = \sum_P \frac{\vec{\Omega}_P(t) \times \vec{r}_P}{4\pi r^3}(1 - \exp(r^3/\varepsilon^3)) \tag{25}$$

where $\vec{r}_P = \vec{x} - \vec{Z}_P$. $\vec{\Omega}_P$ and $\vec{Z}_P$ are determined by integrating in time the following evolution equations:

$$\frac{d\vec{Z}_P(t)}{dt} = \vec{u}(\vec{Z}_P;t), \quad \frac{d\vec{\Omega}_P(t)}{dt} = \left(\vec{\Omega}_P(t)\nabla\right)\vec{u}(\vec{Z}_P;t) \tag{26}$$

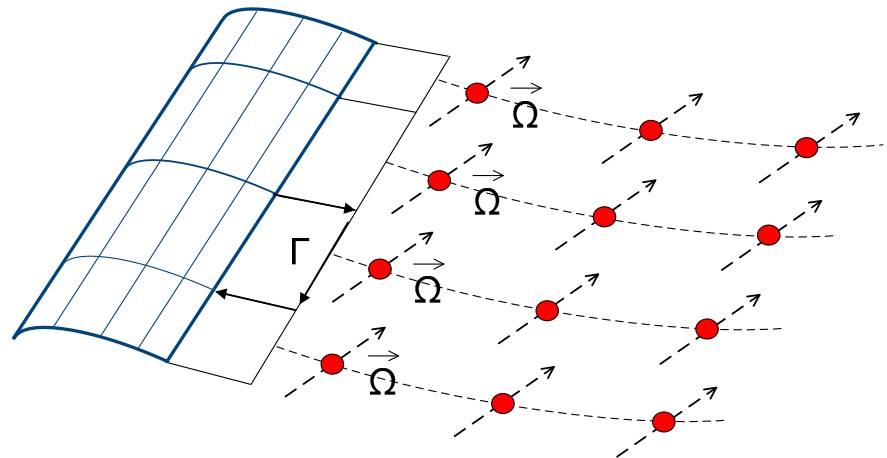

**Figure 8.** Wake representation by free vortex particles.

Being unconnected, the vorticity field defined by the vortex blobs will be distorted and numerical errors will accumulate. In order to correct this, remeshing is applied, which, besides the improvement in accuracy, also makes it possible to apply the particle-mesh method that substantially reduces the computational cost. Especially when the implementation makes use of the advantages parallel processing offers, it is possible to perform complete 10 min simulations that will involve several millions of particles. In the particle mesh method, free (space) vorticity is first projected on a Cartesian grid and then the Poisson equation $\nabla^2\vec{\psi} = -\vec{\omega}$ is solved over the same grid by means of a fast Fourier transformation solver [35]. Then $\vec{u}_{wake} = \nabla \times \vec{\psi}$ and its derivatives appearing in (26) are calculated through finite differences.

### 2.3. Hydrodynamic Modeling

2.3.1. Potential Flow Modeling

In the time domain, the equation of motion of a rigid floater takes the following form:

$$(\mathbf{M} + \mathbf{a}_\infty)\ddot{\mathbf{q}}_f + \int_0^t \mathbf{R}(t-\tau)\dot{\mathbf{q}}_f(\tau)d\tau + (\mathbf{K}_H + \mathbf{K}_G + \mathbf{K}_{Moor})\mathbf{q}_f = \mathbf{F}_{exc}^{(1)} + \mathbf{F}_{exc}^{(2)} + \mathbf{F}_{Moor} + \mathbf{F}_{WT} + \mathbf{F}_{visc} + \delta_{l3}(B-W) \quad (27)$$

where vector $\mathbf{q}_f$ contains the three displacements and three rotations of the floater; dots denote time derivatives; $\mathbf{M}$ is the $6 \times 6$ generalized mass matrix of the floater; $\mathbf{a}_\infty$ denotes the added mass matrix corresponding to infinite wave frequency; $\mathbf{R}(t)$ denotes the retardation matrix in the time convolution that captures linear hydrodynamic damping due to memory effects and is defined by,

$$R_{ij}(t) = \frac{2}{\pi}\int_0^\infty b_{ij}(\omega)\cos(\omega t)d\omega \quad (28)$$

where $b_{ij}$ denote the elements of the added damping matrix $\mathbf{b}$; $\mathbf{K}_H$, $\mathbf{K}_G$ and $\mathbf{K}_{Moor}$ denote the stiffness matrices due to hydrostatics, gravity and mooring lines (zero if the dynamic mooring line model is used); $\mathbf{F}_{exc}^{(1)}$ denotes the first –order wave exciting force accounting for the wave force and diffraction, and is defined as follows:

$$\mathbf{F}_{exc}^{(1)}(t) = A\left[\frac{\mathbf{F}_{exc}^{(1)}(\omega)}{A}\right]\cos(\varphi(\omega)-\omega t)$$
$$\mathbf{F}_{exc}^{(1)}(t) = \sum_{i=1}^n A_i\left[\frac{\mathbf{F}_{exc}^{(1)}(\omega_i)}{A}\right]\cos(\varphi_i(\omega_i)-\omega_i t + \varepsilon_i),\ A_i = \sqrt{2S(\omega_i)d\omega} \quad (29)$$

where the first expression in (29) is valid for a single Airy wave of height 2A, while the second expression is valid for irregular waves described by a spectrum with spectral density $S(\omega)$. The term in square brackets and $\varphi(\omega)$ correspond to the magnitude (per wave amplitude) and the phase angle of the exciting force calculated in the frequency domain, while $\varepsilon_i$ is the random phase angle of every ith wave component, that is uniformly distributed over $[0, 2\pi]$; $\mathbf{F}_{exc}^{(2)}$ in (27) denotes the second-order wave-induced force vector that is approximated using Newman's approximation [13]:

$$\mathbf{F}_{exc}^{(2)}(t) = \left[\sum_{i=1}^n A_i\sqrt{2\left[\frac{\mathbf{F}_{drift}(\omega_i)}{A^2}\right]}\cos(-\omega_i t + \varepsilon_i)\right]_{\mathbf{F}_{drift}(\omega_i)>0}^2$$
$$-\left[\sum_{i=1}^n A_i\sqrt{-2\left[\frac{\mathbf{F}_{drift}(\omega_i)}{A^2}\right]}\cos(-\omega_i t + \varepsilon_i)\right]_{\mathbf{F}_{drift}(\omega_i)<0}^2,\ A_i = \sqrt{2S(\omega_i)d\omega} \quad (30)$$

where $\mathbf{F}_{drift}$ is the mean drift force.

In the above, the added mass and damping matrices, the exciting force and the mean drift force vectors are calculated a priori by solving the first-order hydrodynamic problem in the frequency domain using panel solvers such as WAMIT [48] or freFLOW [49,50].

2.3.2. Morison's Equation

According to Morison's equation [15], the hydrodynamic force per unit length L of an inclined, submerged and moving body takes the following form:

$$dF/dL = \underbrace{\rho\,dV\,a_n}_{\text{Froude krylov}} + \underbrace{C_a\,\rho\,dV\,a_n}_{\text{Diffraction}} - \underbrace{C_a\,\rho\,dV\,\ddot{q}_n}_{\text{Added mass}} + \underbrace{0.5\,C_d\,\rho\,dS\,|u-\dot{q}|_n(u-\dot{q})_n}_{\text{Drag term}} \quad (31)$$

where a and u denote the total (wave and current) induced acceleration and velocity, respectively, which are derived from Airy's or stream function wave theory, while $\ddot{q}$, $\dot{q}$ denote the acceleration and the velocity of the body. Subscript n denotes the normal to the surface component. $C_a$ and $C_d$ are the added mass and drag coefficients, respectively. dV and dS denote the infinitesimal volume and the surface normal to the flow, respectively (i.e., for a cylinder of radius R, $dV = \pi RP^2 PdL$ and $dS = 2RdL$). $\rho$ denotes the density of the water. In the above expression, the total force is the sum of the Froude Krylov force, the diffraction force, the added mass force and the drag force.

The Froude Krylov force, as considered in Morison's equation, is valid only for fully submerged bodies, as reported, for example, in [11]. In the surface piercing (floating) case, the Froude Krylov force on the end nodes should be expressed in terms of the dynamic pressure $p_d$ over the wetted surface.

### 2.3.3. Hydrostatics

In case the floater is rigid, the total buoyancy (concentrated load) and the linearized hydrostatic stiffness terms are directly considered in the equation of motion (see Equation (27)). On the contrary, if the substructure is flexible, the hydrostatic pressure is integrated along the flexible members. When conical members are used, pressure integration over their wet surface leads to the following expressions (defined in local coordinates):

$$d\mathbf{F}_b/dL = \left\{ \begin{array}{c} \rho g S T_{31} \cos\theta_c \\ -P\left(\rho g z_g + p_d\right)\sin\theta_c \\ \rho g S T_{33} \cos\theta_c \end{array} \right\} dL, \quad d\mathbf{M}_b/dL = \left\{ \begin{array}{c} \rho g S R T_{33} \sin\theta_c \\ 0 \\ -\rho g S R T_{33} \sin\theta_c \end{array} \right\} dL$$

$$F_{bn} = \rho g z_g S$$

(32)

where the first two terms (distributed force and moment per unit length) are applied on the side surfaces, while the last term (concentrated force) is applied at the end nodes (e.g., bottom of the floater). In the above equations, S, P and R denote the area, the circumference and the radius of the member at the point where the buoyancy force is evaluated, respectively, $z_g$ the global vertical coordinate of the center of the section, $p_d$ the dynamic pressure, $\theta_c$ the cone angle of the member and $T_{ij}$ the elements of the local to global transformation matrix. In the present convection, the y-axis in the local system coincides with the beam axis.

It is noted that, by linearizing the rotation matrix **T** and the global position vector $z_g$ in (32), convergence is significantly improved, whereas hydrostatic stiffness terms are inherently introduced in the equation of motion of the floater.

### 2.4. Wind and Wave Excitation

Wind inflow is specified by defining its three velocity components (u, v, w) over a space grid (wind box). The wind field is decomposed into its deterministic and stochastic parts. The deterministic part may vary in space (e.g., due to shear and topography) and time (e.g., in the case of a deterministic wind gust). For the stochastic part, time-series of the wind fluctuations u', v', w', in all three directions in space are determined. These series are obtained by specifying the power spectral density (PSD) function $S(\omega)$ of all wind components and a spatial correlation function. The standard choice is the von Karman spectrum, combined with Davenport's exponential coherence function [51]. Time series of turbulent fluctuations are generated over the inlet cross section of the wind (in the yz plane), by combining inverse Fast Fourier Transform (FFT) and sampling of distributed random phases (method proposed by Veers' [52]). By applying Taylor's hypothesis, which postulates that disturbances travel with the average wind velocity $U_W$, time dependence is transformed into x dependence; $x = t\ U_W$, where x is in the wind prevailing direction and so turbulent velocity fluctuations are defined over the so called "turbulent wind box".

Incoming waves are generated in a similar approach. The spectrum concerns the energy of the wave and is associated with the wave height. Pierson–Moskowitz or JONSWAP spectra [11]

are usually considered, but also other site-specific ones may be used. The shape of the spectrum is parametrically defined by the peak period $T_P$ and the significant wave height $H_S$, as well as additional specific shape parameters depending on the spectrum considered (see [11]).

For stochastic wave excitation, wave kinematics is almost exclusively based on linear Airy theory and the wave field is defined as a superposition of regular sinusoidal waves. The surface elevation of an irregular sea state described by a $S(\omega)$ is defined as follows:

$$\zeta(x; t) = \sum_{i=1}^{n} A_i \cos(k_i x - \omega_i t + \varepsilon_i), A_i = \sqrt{2S(\omega_i)d\omega} \tag{33}$$

where $\varepsilon_i$ and $k_i$ denote the random phase angle uniformly distributed over $[0, 2\pi]$ and the wave number, respectively, of the ith wave component. A is the wave amplitude and x is the space coordinate along which $\zeta$ is evaluated. It is noted that, in Airy theory, wave kinematics is, by definition, valid up to the mean sea water level (MSL). When using Morison's equation, extrapolation methods may apply to extend the application range of Airy theory up to the instantaneous water level (IWL). Amongst the various engineering extrapolation methods, Wheeler stretching [53] is here considered.

*2.5. Controller*

A built-in baseline generic control module may be considered, which simulates a standard power-speed controller (variable speed/variable pitch), designed to maximize power output in partial load operation and to regulate power/rotor speed in above rated operation, respectively. In the present case, standard, linear Proportional Integral Differential (PID) control elements and linear filters are used. Start-up and shut-down modes are also available in order to fulfill the requirement of the standards for simulating start and stop sequences. Additional control logic is implemented in the baseline controller that accounts for individual pitch and/or flap control for load reduction [54–56], thrust peak shaving near rated wind speed for load mitigation (standard option in modern commercial turbines) and soft shut down/storm control option to delay the shut-down of the turbine above the standard cut-out wind speed (i.e., 25 m/s). Alternatively, it is possible to plug-in in the simulation external control libraries, which solve internally the control equations and provide control variables.

The controller interacts with the structural dynamic module at the generator side drive-train node, where input generator torque is applied, and at the root of the blades, where pitch motion is imposed.

## 3. Numerical Results and Discussion

The numerical results next presented, concern the NREL 5MW reference wind turbine (WT) [57] of the Offshore Code Comparison Collaboration Continuation (OC4) phase II and the Offshore Code Comparison Collaboration (OC3) phase IV IEA Annexes, in which it was mounted on a semi-submersible [58] and a spar-buoy floater [59], respectively. The baseline NREL controller [57] is employed, which implements a variable speed/variable pitch power controller.

In Section 3.1, emphasis is given on the modeling of the floating support structure and the mooring lines, as well as on the verification of the coupled methodology. Predictions are compared against data of state-of-the-art numerical tools that participated in OC4 project in 2015 [37]. Both available hydrodynamic modeling options (i.e., potential theory and Morison's equation) are employed, alongside with the flexibility of the floater, although the floater is rather stiff. In Section 3.2, a challenging case regarding aerodynamics is presented, in which the operation of a floating wind turbine in half-wake condition is considered. The free-wake vortex particle method, medium fidelity aerodynamic option is employed in order to study one- and two-way wake interactions, as well as to calibrate BEMT.

*3.1. Verification of the Hydro-Servo-Aero-Elastic Modeling*

In this case the WT was mounted on a steel semi-submersible floater at a depth of 200 m. Four codes were selected for comparisons, out of the list of the participating codes in OC4 phase II, as shown in

Table 1. All codes use BEMT aerodynamic modeling. The present (hGAST), HAWC2 and Orcaflex are FEM-based codes and also adopt dynamic mooring line modeling, while BLADED and FAST are modal-based codes with quasi static mooring line modeling. HAWC2 and BLADED use Morison's equation for the hydrodynamic modeling, while FAST and Orcaflex use potential theory.

As mentioned, both hydrodynamic models are implemented in present code, denoted hereafter as hGASTp and hGASTm, to distinguish potential and Morison-based application, respectively. Wave kinematics in Morison's equation is calculated in the instantaneous position (IP) of the body and up to the instantaneous water level (IWL) using Wheeler stretching extrapolation method. Moreover, the Froude Krylov force in Morison's equation at the end nodes is expressed in terms of the dynamic pressure, while hydrostatics (i.e., buoyancy force and restoring moments) is calculated in nonlinear manner using the surface integration method (see Section 2.3). It is noted that, in the OC4 project, the considered hydrodynamic coefficients in Morison's equation were calibrated on the basis of the linear potential theory to allow consistent comparison between Morison and potential based code predictions, while additional damping was introduced in the potential based tools (i.e., drag term in Morison's equation or external damping terms) to account for viscous drag effects [58].

At first, an overall verification of the dynamic modeling is performed by comparing the natural frequencies of the coupled semi-submersible FOWT system in Table 2. Gravity and structural damping contributions are considered. In general, the agreement is good, indicating consistent modeling of the WT and of the semi-submersible floater. It also proves that the dynamics of the floating system are well captured. Rigid body modes of the floater are similar. The Morison's equation version of the present code and HAWC2 predict slightly reduced surge and sway frequencies of 0.0088 Hz, compared to 0.0093 Hz predicted by the other codes, including the potential version of the present code. Both versions of the present code predict slightly bigger frequencies for the roll and pitch rigid body motions of the floater in comparison to the others.

As regards the WT frequencies, Orcaflex predicts higher tower bending and drive train torsional frequencies. First P blade modes are consistent in the majority of the codes, except that HAWC2 predicts lower blade asymmetric pitch and yaw frequencies in the edgewise direction. Differences between modal and FEM codes are found in the second tower fore-aft and side-to-side bending modes and the second blade asymmetric yaw mode in the flapwise direction. Modal codes overestimate the frequency of the aforementioned modes. For example, FAST over predicts the second tower side-to-side bending mode at 5 Hz, while the other codes predict it at ~3.5 Hz. In addition, both modal codes overestimate the second blade asymmetric yaw mode in the flapwise direction, giving ~1.9 Hz, instead of 1.68 Hz predicted by hGAST and Orcaflex and 1.61 Hz predicted by HAWC2.

The effect of the different hydrodynamic modeling on the natural frequencies is identified by comparing the two different approaches of the present model. WT frequencies remain unchained, while small differences are found in the six rigid body modes of the floater. In the present case, application of Morison's equation gives slightly lower frequencies for the six floater motions, which are in agreement with the HAWC2 results. However, since the Morison's equation is subjected to calibration, a firm conclusion cannot be drawn.

Next, time domain simulation results are presented with all flexibilities of the structure enabled. Uniform wind speed at 8 m/s and an Airy wave of 6 m height and 10 s period, aligned with wind and *x*-axis, is considered at 200 m depth. The controller is operating in variable speed mode.

**Table 1.** Description of the codes selected for comparison out of the list of the OC4 phase II.

| Participant | 4Subsea, Hvalstad, Norway | Technical University of Denmark (DTU), Roskilde, Denmark | DNV GL, Høvik, Norway | National Renewable Energy Laboratory (NREL), Golden, Colorado, USA | National Technical University of Athens (NTUA), Athens, Greece |
|---|---|---|---|---|---|
| Code Name | Orcaflex | HAWC2 | BLADED | FAST | hGAST |
| Structural Dynamics | FEM | FEM | Modal | Modal | FEM |
| Aerodynamics | BEMT | BEMT | BEMT | BEMT | BEMT |
| Hydrodynamics | Potential | Morison | Morison | Potential | Potential Morison |
| Mooring Lines | Dynamic | Dynamic | Quasi static | Quasi static | Dynamic |
| Control | | | NREL 5MW baseline controller | | |

**Table 2.** Natural frequencies [Hz] comparison of the semi-submersible coupled FOWT of the OC4 phase II.

| Mode Description | hGASTp | hGASTm | HAWC2 | BLADED | FAST | Orcaflex |
|---|---|---|---|---|---|---|
| Platform surge | 0.0093 | 0.0088 | 0.0086 | 0.0094 | 0.0093 | 0.0093 |
| Platform sway | 0.0093 | 0.0088 | 0.0088 | 0.0092 | 0.0093 | 0.0093 |
| Platform heave | 0.0583 | 0.0574 | 0.0573 | 0.0581 | 0.0581 | 0.0556 |
| Platform roll | 0.0413 | 0.0404 | 0.0384 | 0.0397 | 0.0392 | 0.0385 |
| Platform pitch | 0.0413 | 0.0404 | 0.0384 | 0.0397 | 0.0392 | 0.0385 |
| Platform yaw | 0.0131 | 0.0126 | 0.0132 | 0.0136 | 0.0132 | 0.0127 |
| First tower fore-aft | 0.424 | 0.423 | 0.424 | 0.425 | 0.426 | 0.465 |
| First tower side-to-side | 0.415 | 0.414 | 0.415 | 0.417 | 0.418 | 0.458 |
| First drivetrain torsion | 0.622 | 0.622 | 0.608 | 0.623 | 0.628 | 0.672 |
| First blade collective flap | 0.717 | 0.717 | 0.686 | 0.706 | 0.704 | 0.692 |
| First asymmetric flapwise pitch | 0.677 | 0.677 | 0.618 | 0.641 | 0.670 | 0.664 |
| First asymmetric flapwise yaw | 0.639 | 0.639 | 0.648 | 0.670 | 0.667 | 0.635 |
| First asymmetric edgewise pitch | 1.079 | 1.079 | 1.002 | 1.080 | 1.079 | 1.095 |
| First asymmetric edgewise yaw | 1.092 | 1.092 | 1.015 | 1.091 | 1.092 | 1.103 |
| Second tower fore-aft | 3.417 | 3.415 | 3.314 | 3.864 | 3.898 | 3.405 |
| Second tower side-to-side | 3.540 | 3.537 | 3.494 | 3.437 | 5.012 | 3.875 |
| Second collective flap | 2.000 | 2.000 | 1.840 | 1.972 | 2.023 | 1.928 |
| Second asymmetric flapwise pitch | 1.876 | 1.876 | 1.739 | 1.718 | 1.914 | 1.829 |
| Second asymmetric flapwise yaw | 1.681 | 1.681 | 1.609 | 1.870 | 1.934 | 1.672 |

In Figure 9, the rigid body motions of the floater are presented. In general, the agreement is good amongst the codes compared. The wave period of 10 s is well represented in all signals. At the 0° wave angle, surge, heave and pitch motions are directly excited by the wave, while sway, roll and yaw motions are only excited due to couplings and; therefore, attain much smaller amplitudes. FAST and the potential version of the present code, that both apply first P order potential theory, predict almost identical surge motion. Small differences in the surge mean value could be attributed to nonlinear hydrodynamic effects [37] or to different aerodynamic thrust force. In the Morison's equation approach, drift effects are caused when the hydrodynamic loads are applied at the instantaneous position (IP) of the body and/or at the instantaneous water level (IWL) using Wheeler's stretching. This is clearly depicted by comparing hGAST potential (solid blue line) and Morison-based predictions (dashed blue line). Morison's equation predicts a negative drift effect in the surge motion. The mean surge is 0.3 m lower as compared to the results based on potential theory that does not include drift effects.

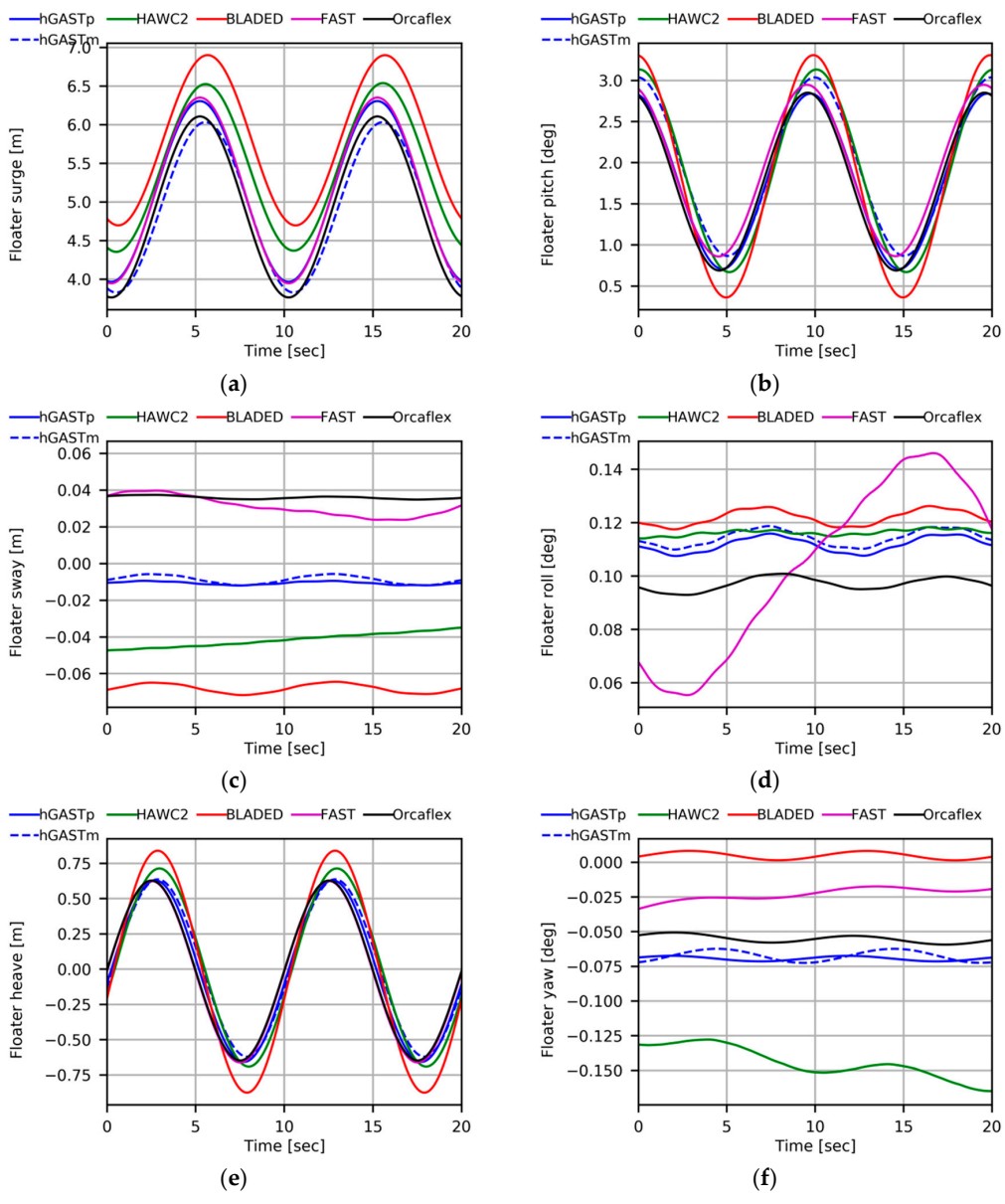

**Figure 9.** Comparison of platform motions: (**a**) Platform surge motion, (**b**) platform sway motion, (**c**) platform heave motion, (**d**) platform pitch motion, (**e**) platform roll motion and (**f**) platform yaw motion (Airy wave *H* = 6 m, *T* = 10 s, uniform inflow at 8 m/s).

The pitch signals are very similar and the mean value, which is determined by the aerodynamic thrust, is consistent. The higher amplitude motion (especially in heave and pitch motions, but also on loading signals in fore-aft direction) predicted by some Morison-based codes could be probably reduced if the dynamic pressure modification discussed in Section 2.3.2 was applied. A small phase, lagging between the potential and Morison-based predictions by the present code, appears because Morison's equation is applied at the instantaneous position, while potential theory at the undisturbed, reference position of the floater. A minor increase in the pitch mean value by ~0.2° is also predicted by Morison's equation.

Although sway, roll and yaw motions attain small values, the agreement is very good. These motions are less damped, so, in certain cases, the provided results have not completely reached a periodic state. The positive roll angle is set by the torque of the rotor. Differences in the sway sign are not straightforward to explain, because of the very small values attained. Heave motion is well represented. Again some of the codes using Morison's equation predict higher amplitudes.

In Figure 10a, the tension of the mooring line at fairlead 2 (upwind position) is compared. The mean values are identical, while phase shift and higher frequencies appear in the codes that adopt the dynamic mooring lines modeling, as compared to those that adopt the quasi static approach. These differences do not influence the dynamics of the coupled FOWT system (see Figure 9), but will affect the Damage Equivalent Loads (DELs) of the mooring lines [37], which could be of crucial importance in the design of the mooring lines and the foundation.

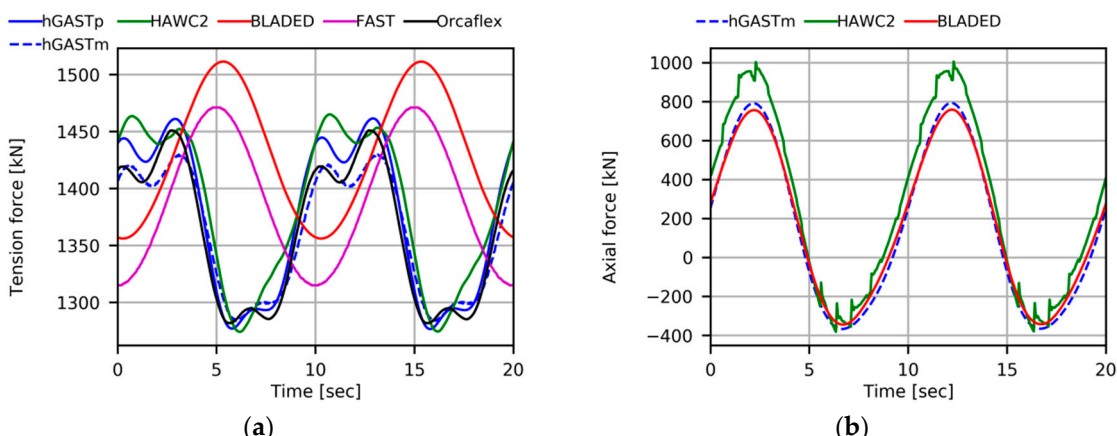

**Figure 10.** Comparison of (**a**) mooring line tension at fairleads 2 (upwind) position and (**b**) local axial force of the upper horizontal member of the floater at the connection point between column 1 and delta connection (Airy Wave *H* = 6 m, *T* = 10 s, uniform inflow at 8 m/s).

Figure 10b presents the local axial force of the upper horizontal bracer of the floater at the connection point between the first vertical column and the upper delta connection. Only Morison-based codes that consider the flexibility of the floater (through 1D beam modeling) provided results for this signal. Good agreement is noted, whereas the wave period of 10 s is again depicted in the loading signal.

In Figure 11, the fore-aft and yawing moments at the tower base are compared. The wave period is depicted in both tower signals. Signals in the fore-aft direction are mainly affected by the wave excitation. Higher amplitude predictions in Morison-based codes are explained from the higher amplitudes of the pitch motion. Expected differences in the mean value of the tower yaw moment between FEM and modal-based codes are masked by the increase in amplitude due to the rigid motions of the floater. Additionally, high frequency excitations are predicted by most of the codes, as a result of the lowly damped yaw motion as compared to the fore-aft motion.

In Figure 12, the out-of-plane moment at the root of the blade and the torsion angle at its tip are presented. The rotor period of about 6 s is dominant in both signals. Bending moment in the out-of-plane direction is consistent, although slight differences exist. The out-of-plane signals are

also affected by the presence of the wave, leading to increased amplitudes, as compared to onshore or bottom-fixed supporting structures. Both wave and rotor frequencies are depicted.

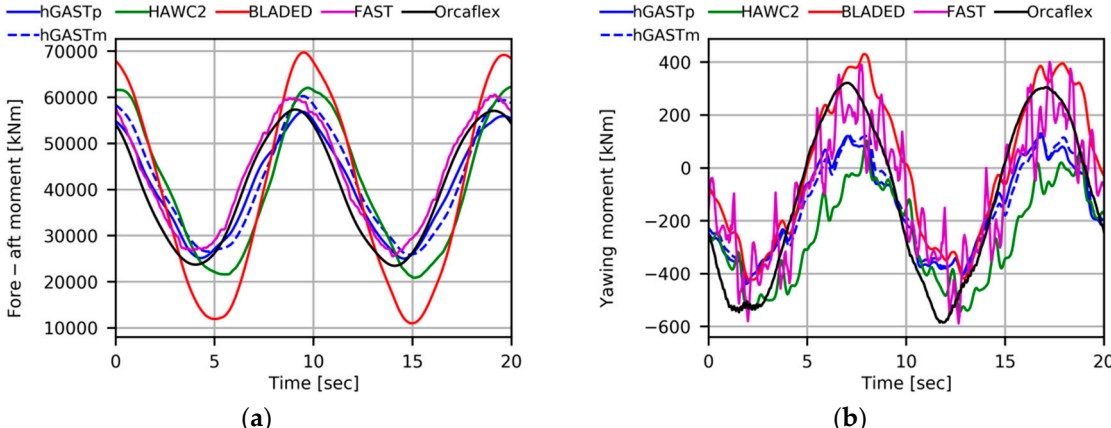

**Figure 11.** Comparison of the tower base (**a**) fore-aft moment and (**b**) yawing moment (Airy wave $H = 6$ m, $T = 10$ s, uniform inflow at 8 m/s).

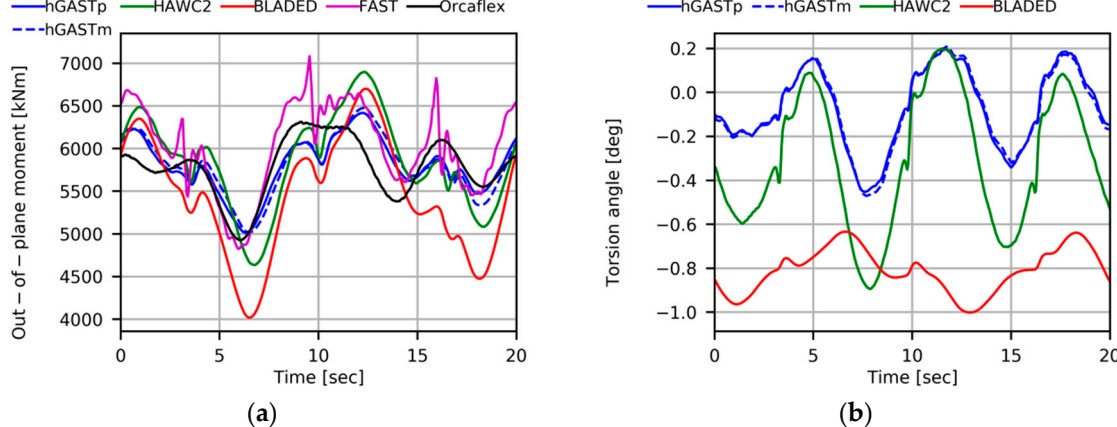

**Figure 12.** Comparison of the (**a**) blade root out-of-plane bending moment and (**b**) blade tip torsion angle (Airy wave $H = 6$ m, $T = 10$ s, uniform inflow at 8 m/s).

Differences are seen in the torsion angle. The present model and HAWC2, that both account for geometric nonlinear effects due to large deflections and rotations, predict similar results with the same phase, but with different mean value and amplitude [18]. In BLADED results, the difference in phase and the smaller amplitude is possibly explained by the use of a simplified beam model.

Out of the above comparisons, it is concluded that the present model (hGAST) is thoroughly verified through code-to-code comparisons against state-of-the-art tools participated in IEA OC4 Annex. Moreover, both potential and Morison-based hydrodynamic modules may provide similar predictions in the case that Morison's equation is properly tuned. Morison's equation models can be adopted in cases where the flexibility of the floater is considered. Finally, phase shift and higher frequencies appeared in the mooring line tension in the codes that adopt the dynamic mooring lines modeling, as compared to those that adopt the quasi static approach. These differences are not found to influence the dynamics of the coupled FOWT system but could be critical in the design of the mooring lines and the foundation.

### 3.2. Half-Wake Effects on Floating Offshore Wind Turbines (FOWT)

The present test case concerns two floating wind turbines placed at a distance of 5 diameters (D) along the prevailing wind direction with an off-set of 1 radius (0.5 D) in the lateral direction, so half of the rotor of the second FOWT is submerged into the wake of the first. A simplification that BEMT aerodynamic cannot avoid, assumes that there is one-way interaction, meaning that only the wake of the preceding WT affects the performance of the second one. In theory the interaction is two ways, in the sense that the incoming wake is modified by the presence of the second WT. A similar two-way interaction also takes place when inflow turbulence impinges a WT. In both cases, the incoming disturbance can be interpreted as a connection of free vortices, which renders vortex methods a natural choice.

An illustration of how vortex methods can be applied in this half-wake interaction problem is shown in Figure 13. The two WTs are represented by the grid defined on the blades (thin lifting surfaces), while the wakes are depicted by an iso-vorticity surface in scatter data format. In the upwind wind turbine, the formation of the tip and root vortices is clearly represented over the early stages of the wake development, while in the downwind wind turbine the mixing of the two wakes is substantial and does not allow distinguishing well shaped vortex structures. In the figure the front view of the wake development is also shown, indicating the way the preceding wake covers the downwind rotor, as well as the expansion of the wake.

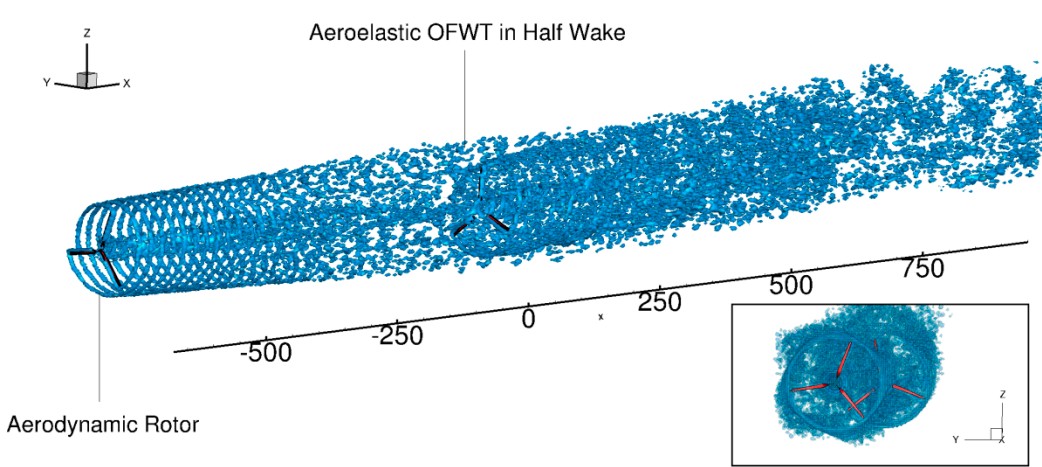

**Figure 13.** Snapshot of the wake formation in the case of two interacting wind turbines (WTs). An iso-vorticity surface is used to mark the formed wake structure.

Quantitative information is given by the instantaneous and averaged normalized deficit velocity profiles at 5 D and 8 D in the wake of an isolated WT, that are shown in Figure 14. The velocity deficit attains a mean value of about 35% at 5 D and 30% at 8 D, respectively. The dots in the figure represent the raw instantaneous data over the last period obtained from the simulation every time step, while the lines give the corresponding average, which is used as input in the one-way interaction model simulations.

The simulations performed are listed in Table 3 and concern the NREL 5MW Reference WT [57] mounted on the OC3 spar-buoy floater [59]. In the numerical simulations, only the second WT is flexible. Uniform wind inflow (no ambient turbulence, yaw and shear) is chosen at the rated wind speed of 11.4 m/s, while, following the definition of the OC3 project, a regular wave of $H = 6$ m and $T = 10$ s is considered at the water depth of 320 m. Rated conditions have been chosen, because for a variable pitch, variable speed WT, maximum loading is obtained leading to highest momentum deficit in the wake. The list of the load cases concerns the isolated case in uniform inflow conditions (1 rotor), the one-way interaction modeling or frozen wake at x = 5 D and the two-way interaction modeling (2 rotors) performed either using the BEMT or the vortex free-wake aerodynamic method.

Next, these modeling options are compared in terms of time series and power spectral densities (PSDs) for selected signals recorded on the second (hydro-aero-elastic) WT.

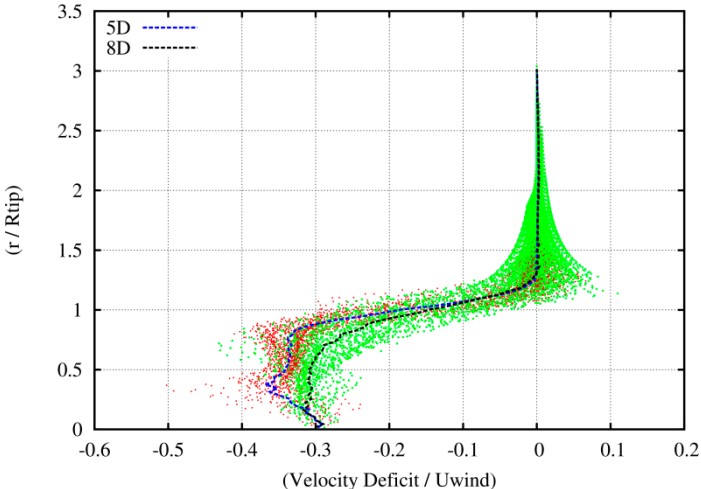

**Figure 14.** Instantaneous (dots) and time averaged (dash lines) axial velocity deficits at x = 5 D and 8 D. The green and red dots correspond to downwind positions of 5 D and 8 D respectively.

**Table 3.** Load cases at rated wind speed of 11.4 m/s.

| Description of Load Case | Vortex | BEMT |
|---|---|---|
| Isolated WT: uniform inflow, no wave excitation, pure aerodynamic simulation | Yes | No |
| Isolated WT: uniform inflow, wave excitation and flexibility included | Yes | Yes |
| WT in half-wake operation: one-way interaction wake model (the impacted WT is only simulated) | Yes | Yes |
| WT in half-wake operation: two-way interaction wake model (both WTs are included in the simulation) | Yes | No |

In Figure 15a, time series of 200 s long are presented for the rotor speed. The excitation by the wave of 10 s period is clearly seen. In the two-way interaction simulation, at about $t = 55$ s the wake of the upwind rotor has covered the 5 D distance. The interaction leads to a slight reduction of the mean value of the rotor speed which becomes almost identical to the vortex based one-way interaction simulation (black solid and dashed light blue lines). In both BEMT sets of results, the mean rotor speed obtained is 3% lower than that vortex models predict.

In Figure 15b, time series of 200 s long are presented for the yaw angle of the floater. As previously, transition from uniform to half-wake inflow conditions starts at about $t = 55$ s in the two-way results. Before, the yaw motion is the same as in the isolated case, whereas, when the transient dies out at about $t = 100$ s, the motion converges to that of the one-way interaction model with minor deviation in amplitude. In general, BEMT and vortex aerodynamic predictions agree well. It is noted that the yaw motion is chosen here because the wave dominates the rest of the motions, so a clear effect of the half-wake interaction is only visible in the yaw motion due to the loading asymmetry (wave excitation in yaw motion of the spar-buoy floater is zero).

Next, focusing on WT loading, time series of the fore-aft bending moment at the tower base are presented in Figure 16a. Since the half-wake effect is masked by the wave loading, the mean value and the amplitude remain almost unchanged, although high excitation frequencies appear. In order to identify them, in Figure 16b the PSD of the tower base fore-aft bending moment is presented. Clearly the wave frequency at 0.1 Hz attains the highest value. However, in both the one- and two-way

interaction models, the 3P (three per revolution) frequency at 0.6 Hz is also excited, as well as the tower natural frequency at 0.45 Hz. The latter is found to be more excited in the two-way model, as shown in both plots.

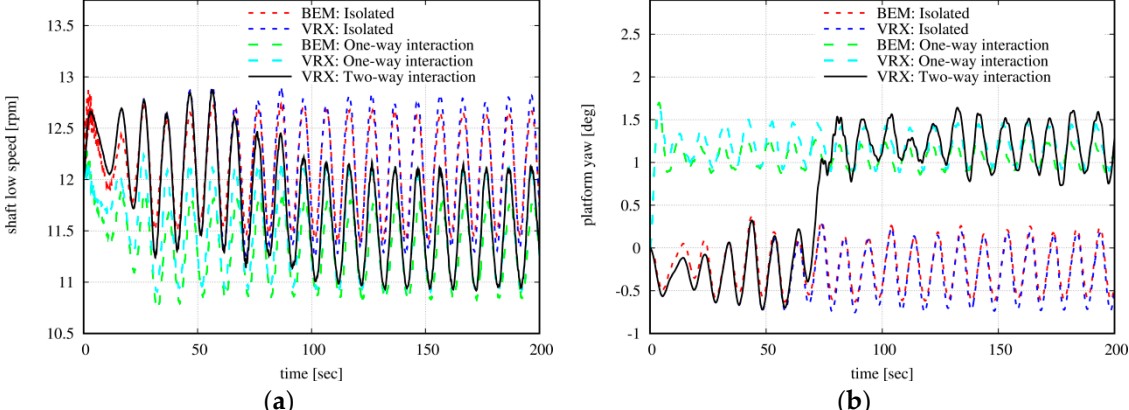

(a)    (b)

**Figure 15.** Time-series of the (**a**) rotational speed and (**b**) yaw motion of the floater. Comparisons between the isolated inflow case and the one- and two-way interaction models for half-wake modeling at x = 5 D using blade element momentum theory (BEMT) and Vortex (VRX) aerodynamic models for the OC3 spar-buoy FOWT at 11.4 m/s.

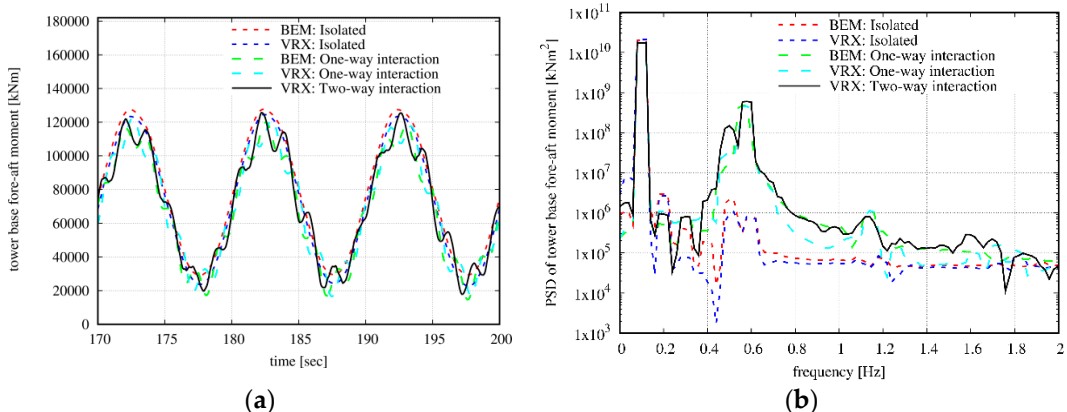

(a)    (b)

**Figure 16.** (**a**) Time-series and (**b**) Power spectral densities (PSD) of the fore-aft bending moment at the tower base. Comparisons between the isolated inflow case and the one- and two-way interaction models for half-wake modeling at x = 5 D using BEMT and Vortex (VRX) aerodynamic models for the OC3 spar-buoy FOWT at 11.4 m/s.

In Figure 17, the tower yaw moment signal is considered. Time averaged values and amplitudes shown in Figure 17a increase when there is half-wake effect, which is responsible for the loading asymmetry on the rotor. In quantitative terms, there is fair agreement. In the interaction case, the BEMT amplitudes are lower (almost half). In Figure 17b, the PSD of the yaw moment is presented. High excitation takes place at 0.1 Hz, which is the wave frequency, but also at 0.6 and 1.2 Hz, which are the 3P and 6P frequencies of the rotor, respectively. In terms of peaks, the agreement is good. However, the excitation in the two-way interaction spectrum is broader, especially at low frequencies. Because the wave excites the yaw direction only indirectly through the couplings with the other floater motions, the most excited frequency is the 3P at 0.6 Hz, also visible in the time series.

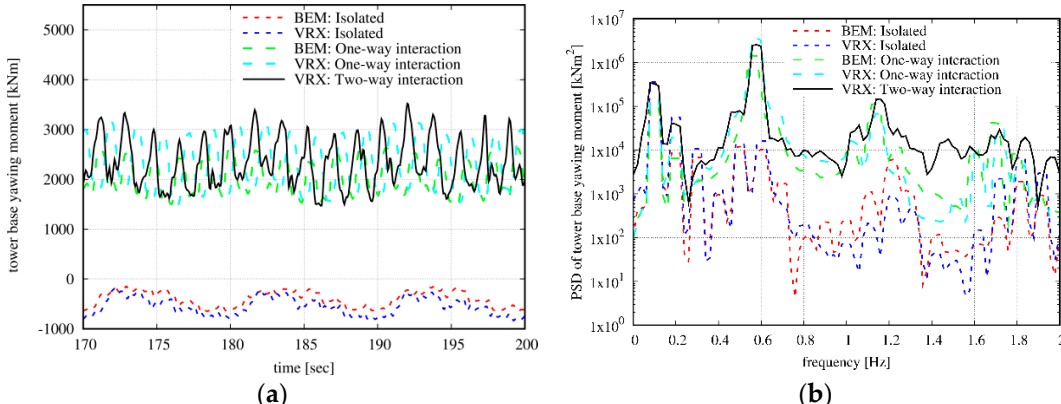

**Figure 17.** (**a**) Time series and (**b**) PSD of the yawing moment at the tower base. Comparisons between the isolated inflow case and the one- and two-way interaction models for half-wake modeling at x = 5 D using BEMT and Vortex (VRX) aerodynamic models for the OC3 spar-buoy FOWT at 11.4 m/s.

Finally, in Figure 18 the signal of the out-of-plane bending moment at the blade root is considered. In the time series, the mean level and the min-to-max amplitude are in fair agreement. However, in their details the variations of the specific load are different. Deviations of this kind are reflected on the fatigue loading. Additionally, phase differences are noted. Because it is not possible to synchronize the rotor speed with the wave elevation, such differences are expected. The PSDs shown on the right indicate the dominance of the wave seen at 0.1 Hz, but also the significance of the blade passage through the incoming wake flow at 0.2 Hz which corresponds to 1P. It is worth noticing that 2P and 3P are also excited in both modeling options of the half-wake interaction case. The agreement between the two interaction models is good.

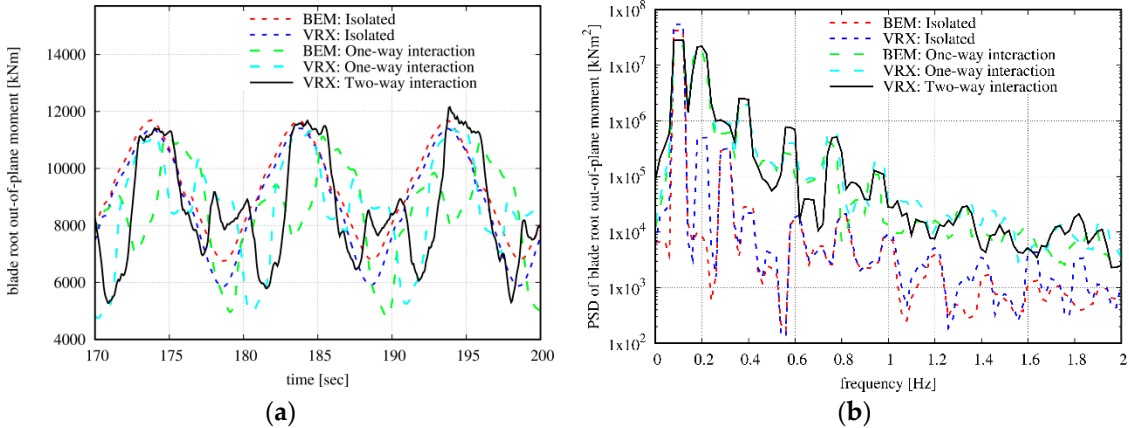

**Figure 18.** (**a**) Time series and (**b**) PSD of the out-of-plane bending moment at the blade root. Comparisons between the isolated inflow case and the one- and two-way interaction models for half-wake modeling at x = 5 D using BEMT and Vortex (VRX) aerodynamic models for the OC3 spar-buoy FOWT at 11.4 m/s.

Out of the above comparisons, it follows that one- and two-way interaction modeling lead to similar results. It is also found that BEMT, which can only model the one-way interaction, is consistent as compared to vortex modeling. This point requires a deeper investigation that should cover a variety of designs and load cases over the full operation span. Deviations appeared in the load ranges which could affect fatigue. However, in order to support this point, proper fatigue simulations are needed.

## 4. Concluding Remarks

An all-inclusive hydro-servo-aero-elastic simulation tool for FOWTs was presented together with indicative results. By following the generalized formulation of analytic dynamics, a simulation environment was defined, in which interfaces realizing kinematic and loading compatibility allow to plug-in different sub-models without canceling any of the couplings or changing the non-linear character of the underlying interactions. In this respect, sub-models can be any of the aerodynamic, hydrodynamic, elasto-dynamic and controlling aspects of the full problem. One advantage of such an approach is that formal linearization reformulates the equations in incremental form, which facilitates the iterative solution of the fully non-linear problem and at the same time formulates the linear stability problem for the full configuration [18].

Targeting holistic analysis of complex systems involves compromises that are linked to the available computing capacity, the time frame and the amount of simulations needed. In the present simulation tool, the main compromise concerns the aerodynamic modeling and the choice amongst the BEMT and the vortex options. The BEMT version was verified via the code-to-code comparisons with other state-of-art BEMT tools for the semi-submersible 5 MW FOWT in Annex OC4 (results in Section 3.1). From the comparisons, it was concluded that both potential and Morison-based hydrodynamic modules may provide similar predictions in the case that Morison's equation is properly calibrated. Additionally, phase shift and higher frequencies appeared in the mooring line tension in the codes that adopt the dynamic mooring lines modeling, as compared to those that adopt the quasi static approach. These differences are not found to influence the dynamics of the coupled FOWT system but could be critical in the design of the mooring lines and the foundation. This justifies the choice of engaging higher order nonlinear submodels in the analysis of the complex FOWT system.

By implementing cost reduction techniques, vortex-based simulations of the certification type were made feasible. This allowed a further verification of BEMT predictions in [2], which concluded that, in terms of equivalent loads, BEMT is conservative in comparison to vortex-based modeling. Even in the complex case of half-wake interaction problem, discussed in Section 3.2, the low-cost BEMT performs well, while the one- and two-way interaction modeling in the free-wake aerodynamic method were found to lead to similar results.

Herein, as well as in previous works, the verification campaigns were limited to specific designs and certain loading conditions, in that further confirmation of the findings obtained so far is needed by comparing vortex and CFD-based predictions with BEMT. Important steps in this direction are to actively include inflow turbulence, as indicated in [60], and to extend the comparisons in other operational conditions that are currently poorly addressed, as, for example, in the occurrence of stall- and vortex-induced vibrations.

**Author Contributions:** Conceptualization, D.I.M., V.A.R. and S.G.V.; methodology, all; software, D.I.M., G.P.P. and V.A.R.; validation, D.I.M. and S.G.V.; formal analysis, all; writing-original draft, D.I.M. and V.A.R.; writing-review and editing, V.A.R. and S.G.V.; supervision, S.G.V. All authors have read and agreed to the published version of the manuscript.

**Funding:** This research was funded by the European projects INNWIND.EU (grant number: FP7-ENERGY-2012-1-2STAGE-308974), AVATAR (grant number: FP7-ENERGY.2013.2.3.1/no. 608396), JABACO (grant number: RFSR-CT-2015-00024) and REFOS (grant number: RFCS-02-2015-709526), and the Greek national projects SYNERGASIA-AVRA and ARISTEIA-POSEIDON. The authors would like to acknowledge the European Union and the Greek secretary for the funding.

**Conflicts of Interest:** The authors declare no conflict of interest.

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
