# Peer review of "Hydro-Servo-Aero-Elastic Analysis of Floating Offshore Wind Turbines"

_fluids, doi:10.3390/fluids5040200_

Round 1
Reviewer 1 Report
Fully coupled simulation including hydro-servo-aero-elastic responses are important to well describe the dynamic behaviors of FOWT. From the manuscript, I found that authors have developed very powerful multiphysics simulation tool of FOWT. Authors also compared the results with the other well known programs such as NREL Fast, DTU HAWC2, etc. However, I could not find special novelties (with respect to accuracy, computational cost, first trial, stability, etc) of the proposed software compared to the existed ones in the industrial point of view. If not, author might clarify the academic contribution of the proposed work. I could not agree its publication in Fluids if both are missing. In addition, during uploading process, all equations were broken. It should be modified during the revision process.
Reviewer 2 Report
The paper about the implementation of an off-shore wind turbine simulation code, can be considered globally satisfying as all the aspects appears clear to the readers and scientifically consistent.
The introduction includes a detailed overview on the state of the art where the mathematical background is well discussed and the choices between different algorithms are justified adeguatly.
No evident lacks can be found on the methods as the validation of the code has been assessed through the comparison with other well documented softwares, any small discrepancies between the simulator results have been properly argued.
My only reticence concerns the novelty that the paper introduces respect the many simulation softwares that already exist. In the paper, even if remarkable and technically consistent, should be explicitly expressed wich are the peculiar characteristics of the code that may improve the state of art showing to the reader why you have found necessary the realization of this software.
Reviewer 3 Report
The paper is well-written, the amount and quality of work is remarkable.
The review of the state of the art is relatively brief, considering the huge body of literature available in the field and its fast evolution, but all in all adequate. I spotted some minor aspects which could be better clarified, discussed later in detail.
Minor corrections are needed to the language. For example:
- Page 2, line 93: "aircrafts" should be "aircraft" (uncountable)
- Ref. 25 contains "Proceedings of" twice
In the introduction, the acronym "BEM" is used for "Beam Element Momentum" (whereas, in other contexts, BEM is used for Boundary Element Method, another method sometimes also used for rotor aerodynamics, thus prone to confusion); its use is common practice, for example, in the rotorcraft scientific and engineering community, from which the wind turbine community inherited the methods. However, it comprises two separate theories:
- the Blade Element theory, in which each blade is treated as a single entity, thus a finite number of blades are considered; in this theory, each blade section's circulation and loads does not affect the inflow at other sections, not to mention other blades.
- the Momentum Theory, in which the overall force generated by the rotor is used to determine the overall inflow (either uniform or heuristically distributed over the disk area); in this latter case, an equivalent, infinite number of blades are considered.
When the so-called BEM theory (often better addressed to as BEMT, or BE-MT) is discussed, sometimes confusion is made about the attribution of specific aspects or features. For example:
- Page 2, line 93: the statement "Vortex models are 3D by construction, with strong coupling along the span 94 which is completely absent in BEM" is correct, in the sense that in BE (Blade Element) theory there is no coupling along the span; interaction is loosely recovered through inflow resulting from MT (Momentum Theory), missing for example the localized strong interaction that occurs close to the tip of the blade.
- Page 2, line 95: the statement "while the assumption of infinite number of blades made by BEM" is misleading: the BE (Blade Element) part of the theory does not assume an infinite number of blades; it is the MT (Momentum Theory) that assumes an infinite number of blades.
As a final remark, I humbly suggest - but it is not a requirement - to remove the word "Holistic" from the title.
Round 2
Reviewer 1 Report
I do not agree its academic novelties. However, I understand that SW development is totally different story, and the comparison results with other current existing SW seems to validate the proposed work well. Considering this and other reviewer's opinions, I recommend its publication.